# Oral transfer of chemical cues, growth proteins and hormones in social insects

Adria C LeBoeuf[1,2]*, Patrice Waridel[3], Colin S Brent[4], Andre N Gonçalves[5,6], Laure Menin[7], Daniel Ortiz[7], Oksana Riba-Grognuz[2], Akiko Koto[8], Zamira G Soares[5,6], Eyal Privman[9], Eric A Miska[6,10,11], Richard Benton[1†]*, Laurent Keller[2†]*

[1]Center for Integrative Genomics, University of Lausanne, Lausanne, Switzerland; [2]Department of Ecology and Evolution, University of Lausanne, Lausanne, Switzerland; [3]Protein Analysis Facility, University of Lausanne, Lausanne, Switzerland; [4]Arid Land Agricultural Research Center, USDA-ARS, Maricopa, United States; [5]Department of Biochemistry and Immunology, Instituto de Ciências Biológicas, Universidade Federal de Minas Gerais, Minas Gerais, Brazil; [6]Gurdon Institute, University of Cambridge, Cambridge, United Kingdom; [7]Institute of Chemical Sciences and Engineering, Ecole Polytechnique Fédérale de Lausanne, Lausanne, Switzerland; [8]The Department of Genetics, Graduate School of Pharmaceutical Sciences, The University of Tokyo, Tokyo, Japan; [9]Department of Evolutionary and Environmental Biology, Institute of Evolution, University of Haifa, Haifa, Israel; [10]Department of Genetics, University of Cambridge, Cambridge, United Kingdom; [11]Wellcome Trust Sanger Institute, Wellcome Trust Genome Campus, Cambridge, United Kingdom

*For correspondence: adria.leboeuf@gmail.com (ACLB); Richard.Benton@unil.ch (RB); laurent.keller@unil.ch (LK)

†These authors contributed equally to this work

Competing interests: The authors declare that no competing interests exist.

**Abstract** Social insects frequently engage in oral fluid exchange – trophallaxis – between adults, and between adults and larvae. Although trophallaxis is widely considered a food-sharing mechanism, we hypothesized that endogenous components of this fluid might underlie a novel means of chemical communication between colony members. Through protein and small-molecule mass spectrometry and RNA sequencing, we found that trophallactic fluid in the ant *Camponotus floridanus* contains a set of specific digestion- and non-digestion related proteins, as well as hydrocarbons, microRNAs, and a key developmental regulator, juvenile hormone. When *C. floridanus* workers' food was supplemented with this hormone, the larvae they reared via trophallaxis were twice as likely to complete metamorphosis and became larger workers. Comparison of trophallactic fluid proteins across social insect species revealed that many are regulators of growth, development and behavioral maturation. These results suggest that trophallaxis plays previously unsuspected roles in communication and enables communal control of colony phenotypes.

DOI: https://doi.org/10.7554/eLife.20375.001

## Introduction

Many fluids shared between individuals of the same species, such as milk or semen, can exert significant physiological effects on recipients (*Poiani, 2006*; *Liu and Kubli, 2003*; *Liu et al., 2014*; *Bernt and Walker, 1999*). While the functions of these fluids are well known in some cases (*Liu and Kubli, 2003*; *Liu et al., 2014*; *Bernt and Walker, 1999*; *Perry et al., 2013*), the role(s) of other socially exchanged fluids (e.g., saliva) are less clear. The context-specific transmission and

**eLife digest** Ants, bees and other social insects live in large colonies where all the individuals work together to gather food, rear young and defend the colony. This level of cooperation requires the insects in the colony to be able to communicate with each other.

Most social insects share fluid mouth-to-mouth with other individuals in their colony. This behavior, called trophallaxis, allows these species to pass around food, both between adults, and between adults and larvae. Trophallaxis therefore creates a network of interactions linking every member of the colony. With this in mind, LeBoeuf et al. investigated whether trophallaxis may also be used by ants to share information relevant to the colony as a form of chemical communication.

The experiments show that in addition to food, carpenter ants also pass small ribonucleic acid (RNA) molecules, chemical signals that help them recognize nestmates, and many proteins that appear to be involved in regulating the growth of ants. LeBoeuf et al. also found that trophallactic fluid contains juvenile hormone, an important regulator of insect growth and development. Adding juvenile hormone to the food that adult ants pass to the larvae made it twice as likely that the larvae would survive to reach adulthood. This indicates that proteins and other molecules transferred mouth-to-mouth over this social network could be used by the ants to regulate how the colony develops.

The next steps following on from this work will be to investigate the roles of the other components of trophallactic fluid, and to examine how individual ants adapt the contents of the fluid in different social and environmental conditions. Another challenge will be to determine how specific components passed to larvae in this way can control their growth and development.
DOI: https://doi.org/10.7554/eLife.20375.002

interindividual confidentiality of socially exchanged fluids raise the possibility that this type of chemical exchange mediates a private means of chemical communication.

Social insects are an interesting group of animals to investigate the potential role of socially exchanged fluids. Colonies of ants, bees and termites are self-organized systems that rely on a set of simple signals to coordinate the development and behavior of individual members (*Bonabeau et al., 1997*). While colony-level phenotypes may arise simply from the independent behavior of individuals with similar response thresholds, many group decisions require communication between members of a colony (*LeBoeuf and Grozinger, 2014*). In ants, three principal means of communication have been described: pheromonal, acoustic, and tactile (*Hölldobler and Wilson, 1990*). Pheromones, produced by a variety of glands, impart diverse information, including nestmate identity and environmental dangers. Acoustic communication, through substrate vibration or the rubbing of specific body parts against one another, often conveys alarm signals. Tactile communication encompasses many behaviors, from allo-grooming and antennation to the grabbing and pulling of another ant's mandibles for recruitment to a new nest site or resource.

Ants, like many social insects and some vertebrates (*Boulay et al., 2000a*; *Greenwald et al., 2015*; *Malcolm and Marten, 1982*; *Wilkinson, 1984*), also exhibit an important behavior called trophallaxis, during which liquid is passed mouth-to-mouth between adults or between adults and juveniles. The primary function of trophallaxis is considered to be the exchange of food, as exemplified by the transfer of nutrients from foragers to nurses and from nurses to larvae (*Wilson and Eisner, 1957*; *Buffin et al., 2009*; *Cassill and Tschinkel, 1995*; *Cassill and Tschinkel, 1996*; *Wheeler, 1918*; *Wheeler, 1986*). The eusocial Hymenopteran forgut has evolved a specialized distensible crop and a restrictive proventriculus (the separation between foregut and midgut) enabling frequent fluid exchange and regulation of resource consumption (*Terra, 1990*; *Eisner and Brown, 1958*; *Lanan et al., 2016*; *Hunt, 1991*). In addition to simple nourishment, trophallaxis can provide information for outgoing foragers about available food sources (*Grüter et al., 2006*; *Farina et al., 2007*).

Trophallaxis also occurs in a number of non-food related contexts, such as reunion with a nestmate after solitary isolation (*Boulay et al., 2000b*), upon microbial infection (*Hamilton et al., 2011*), and in aggression/appeasement interactions (*Liebig et al., 1997*). Furthermore, adult ants have been suggested to use a combination of trophallaxis and allo-grooming to share cuticular

hydrocarbons (CHCs) which are important in providing a specific 'colony odor' (*Boulay et al., 2000a*; *Soroker et al., 1995*), although the presence of CHCs in trophallactic fluid has not been directly demonstrated. Given these food-independent trophallaxis events, and the potential of this behavior to permit both 'private' inter-individual chemical exchange as well as rapid distribution of fluids over the social network of a colony, we tested the hypothesis that trophallaxis serves as an additional means of chemical communication and/or manipulation. To identify the endogenous molecules exchanged during this behavior, we used mass-spectrometry and RNA sequencing to characterize the contents of trophallactic fluid, and identified many growth-related proteins, CHCs, small RNAs, and the insect developmental regulator, juvenile hormone. We also obtained evidence that some components of trophallactic fluid can be modulated by social environment, and may influence larval development.

## Results

### Collection and proteomic analysis of trophallactic fluid

Analysis of the molecules exchanged during trophallaxis necessitated development of a robust method for acquiring trophallactic fluid (TF). We focused on the Florida carpenter ant, *Camponotus floridanus*, which is a large species whose genome has been sequenced (*Bonasio et al., 2010*). We first attempted to collect fluid from unmanipulated pairs of workers engaged in trophallaxis, but it was impossible to predict when trophallaxis would occur, and events were usually too brief to collect the fluid being exchanged. We found that after workers were starved and isolated from their colony, then fed a 25% sucrose solution and promptly reunited with a similarly conditioned nestmate, approximately half of such pairs displayed trophallaxis within the first minute of reunion (as observed previously [*Boulay et al., 2000a*; *Dahbi et al., 1999*]), and were more likely to remain engaged in this behavior for many seconds or even minutes. Under these conditions, it was sometimes possible to collect small quantities of fluid from the visible droplet between their mouthparts (referred to as 'voluntary' samples). However, even under these conditions, trophallaxis was easily interrupted, making this mode of collection extremely low-yield.

To obtain larger amounts of TF and avoid the confounding factors of social isolation and feeding status, we developed a non-lethal method to collect the contents of the crop by lightly-squeezing the abdomen of $CO_2$-anesthetised ants (referred to as 'forced' samples, similar to [*Hamilton et al., 2011*]). This approach yielded a volume of $0.34 \pm 0.27$ μL (mean ± SD) of fluid per ant. To determine whether the 'forced' fluid collected under anesthesia was similar to the fluid collected from ants voluntarily engaged in trophallaxis, and ensure that it was not contaminated with hemolymph or midgut contents, we also collected samples of these fluid sources. To compare the identities and quantities of the different proteins found in each fluid, all samples were analyzed by nanoscale liquid chromatography coupled to tandem mass spectrometry (nano-LC-MS/MS) (*Figure 1A*).

Hierarchical clustering of normalized spectral counts of the proteins identified across our samples revealed high similarity between voluntary and forced TF, but a clear distinction of these fluids from midgut contents or hemolymph (*Figure 1A*, for protein names and IDs see *Figure 1—figure supplement 1*). While the voluntary TF samples contained fewer identified proteins than the forced TF samples – likely due to lower total collected TF volume per analyzed sample (< 1 μL voluntary vs. > 10 μL forced) – the most abundant proteins were present across all samples in both methods of collection (*Figure 1A*). To investigate whether the few differences observed between the voluntary and forced TF in the less abundant proteins might be due to the starvation and/or social isolation conditions used to collect voluntary TF, we isolated groups of 25–30 ants from their respective queens and home colonies for 14 days, with constant access to food and water, and collected TF both directly before and after the period of isolation. Social isolation affected the ratios of proteins in TF, with five of the top 40 proteins in TF becoming significantly less abundant and one more abundant when ants were socially isolated (*Figure 1B*; see *Figure 1—figure supplement 2* for names and IDs). Three of the proteins down-regulated in social isolation were also significantly less abundant in voluntary TF samples (from socially isolated ants) relative to forced TF samples (from within-colony ants, *Figure 1—figure supplement 2*). Together these results provide initial evidence that the composition of this fluid is influenced by social and/or environmental experience of an ant, and support the validity of our methodology to force collect TF.

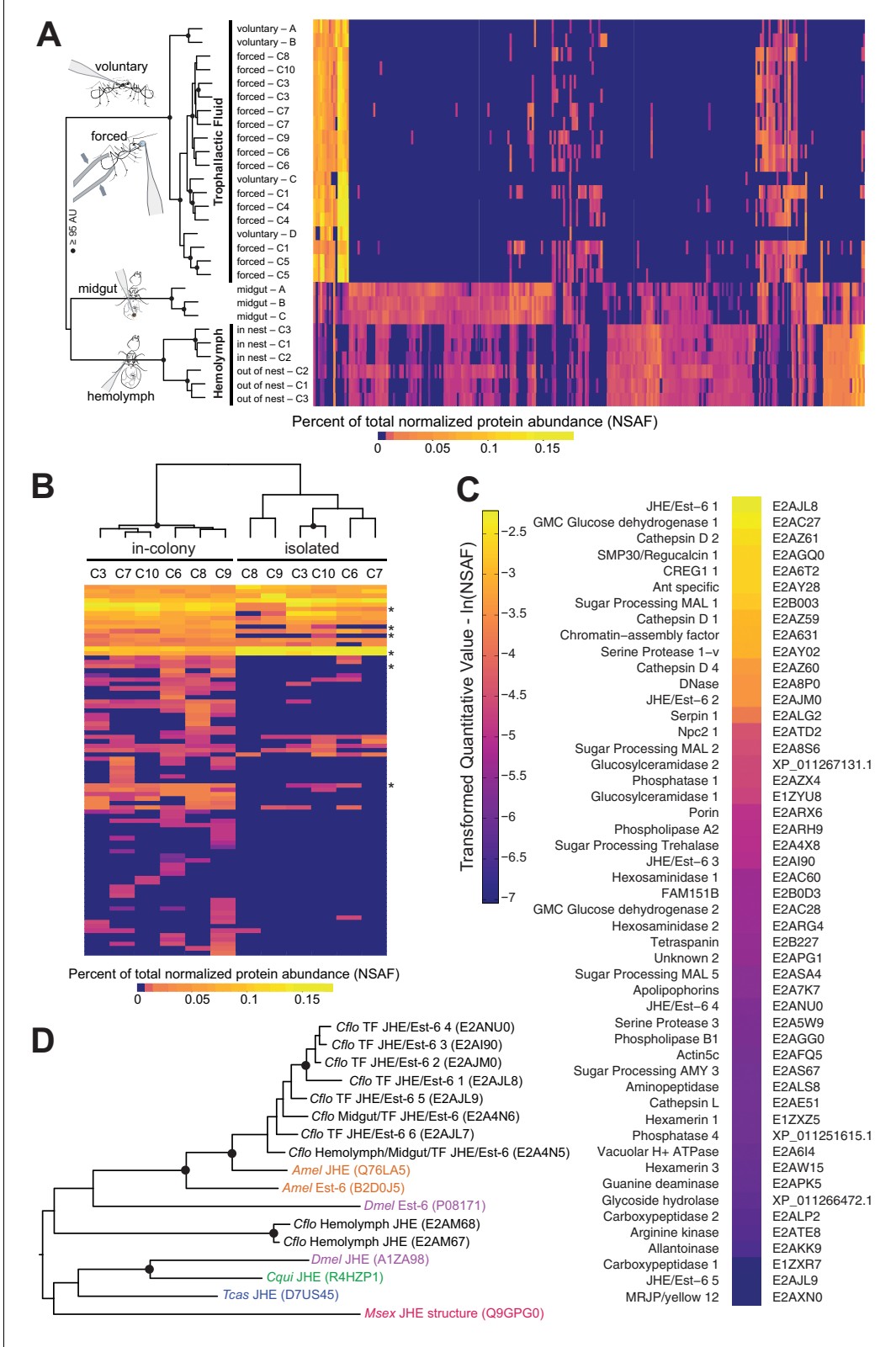

**Figure 1.** Proteomic characterization of *Camponotus floridanus* trophallactic fluid. (**A**) Heat map showing the percentage of total molecular weight-normalized spectra assigned to proteins from voluntary TF, forced TF, midgut, or hemolymph fluids (normalized spectral abundance factor, NSAF [*Zybailov et al., 2006*]). C1-C10 indicate colony of origin. Forced trophallaxis samples are pooled from 10 to 20 ants, hemolymph from 30 ants, and the contents of dissected midguts from five ants each. Voluntary and midgut samples were collected from ants of multiple colonies; multiple samples are

*Figure 1 continued on next page*

*Figure 1 continued*

differentiated by letters. Approximately unbiased (AU) bootstrap probabilities for 10,000 repetitions are indicated by black circles where greater or equal to 95%. (B) Trophallaxis samples from the same ants, in-colony and group-isolated. Trophallactic fluids were sampled first upon removal from the colony, then after 14 days of group isolation (20–30 individuals per group). Values were compared by spectral counting, and the dendrogram shows approximately unbiased probabilities for 10,000 repetitions. Along the right side, asterisks indicate Bonferroni-corrected t-test significance to p < 0.05 between in-colony and isolated TF. Approximately unbiased (AU) bootstrap probabilities for 10,000 repetitions are indicated by black circles where greater or equal to 95%. (C) The most abundant proteins present in TF sorted by natural-log-scaled NSAF value. The UniProt ID or NCBI ID is listed to the right. (D) A dendrogram of proteins including all proteomically observed juvenile hormone esterases (JHE)/Est-6 proteins in *C. floridanus*, the orthologs in *D. melanogaster* and *A. mellifera*, and biochemically characterized JHEs. Each protein name is followed by the UniProt ID. *C. floridanus* JHE/Est-6 proteins are listed with the fluid source where they have been found. Names are color-coded by species. Bootstrap values >95% are indicated with a black circle. JHE/Est-6 6 (E2AJL7) is identified by PEAKS software but not by Scaffold and consequently is not shown in the proteomic quantifications in panels (A–C).

DOI: https://doi.org/10.7554/eLife.20375.003

The following figure supplements are available for figure 1:

**Figure supplement 1.** Proteomic characterization of *C. floridanus* trophallactic fluid.
DOI: https://doi.org/10.7554/eLife.20375.004
**Figure supplement 2.** Social isolation influences trophallactic fluid content.
DOI: https://doi.org/10.7554/eLife.20375.005

Of the 50 most abundant proteins found in TF (*Figure 1C*), many are likely to be digestion-related (e.g. three maltases, one amylase, various proteases, two glucose dehydrogenases, one DNase) and three Cathepsin D homologs might have immune functions (*Hamilton et al., 2011*). However, at least 10 of the other proteins have putative roles in the regulation of growth and development, including two hexamerins (nutrient storage proteins [*Zhou et al., 2007*]), a yellow/major royal jelly protein (MRJP) homolog (likely nutrient storage [*Drapeau et al., 2006*]) and an apolipo-phorin (vitellogenin-domain containing lipid-transport protein [*Kutty et al., 1996*]). Most notably, TF contained several proteins that are homologous to insect juvenile hormone esterases (JHEs) and Est-6 in *Drosophila melanogaster* (*Figure 1D*). JHEs are a class of carboxylesterases that degrade the developmental regulator, juvenile hormone (JH) (*Kamita and Hammock, 2010*; *Nijhout et al., 2014*), and Est-6 is an abundant esterase in *D. melanogaster* seminal fluid (*Mane et al., 1983*; *Richmond et al., 1980*; *Younus et al., 2014*). While most insects have only one or two JHE-like proteins (*Nijhout et al., 2014*; *Qu et al., 2015*), *C. floridanus* has an expansion of more than 10 related proteins, eight of which were detected in TF (*Figure 1D*, *Figure 1—figure supplements 1* and *2*) and two in hemolymph (*Figure 1—figure supplement 1*). Three of the abundant JHE-like proteins are significantly down-regulated in the TF of ants that have undergone social isolation (*Figure 1—figure supplement 2*).

Thirty-three of the 50 most abundant TF proteins had predicted N-terminal signal peptides, suggesting that they can be secreted directly into this fluid by cells lining the lumen or glands connected to the alimentary canal (*Figure 5—figure supplement 1*). Moreover, half of the proteins without such a secretion signal had gene ontology terms indicating extracellular or lipid-particle localization, which suggests that they may gain access to the lumen of the foregut through other transport pathways.

## Trophallaxis microRNAs

Many small RNAs have been found in externally secreted fluids across taxa, such as seminal fluid, saliva, milk and royal jelly (*Sarkies and Miska, 2013*; *Weber et al., 2010*; *Guo et al., 2013*). Although the functions of extracellular RNAs remain unclear (*Turchinovich et al., 2012*), we investigated if TF also contains such molecules by isolating and sequencing small RNAs from *C. floridanus* TF. After filtering out RNA corresponding to potential commensal microorganisms (including the known symbiont, *Blochmannia floridanus*; *Supplementary file 1*) and other organic food components, we detected 64 miRNAs. Forty-six of these were identified based upon their homology to miRNAs of the honey bee *Apis mellifera* (*Guo et al., 2013*; *Søvik et al., 2015*; *Greenberg et al., 2012*), while 18 sequences (bearing the structural stem-loop hallmarks of miRNA transcripts) were specific to *C. floridanus* (*Figure 2*, Figure 2—source data 1). The most abundant of the 64 miRNAs was miR-750, followed by three *C. floridanus*-specific microRNAs. The role of miR-750 is unknown,

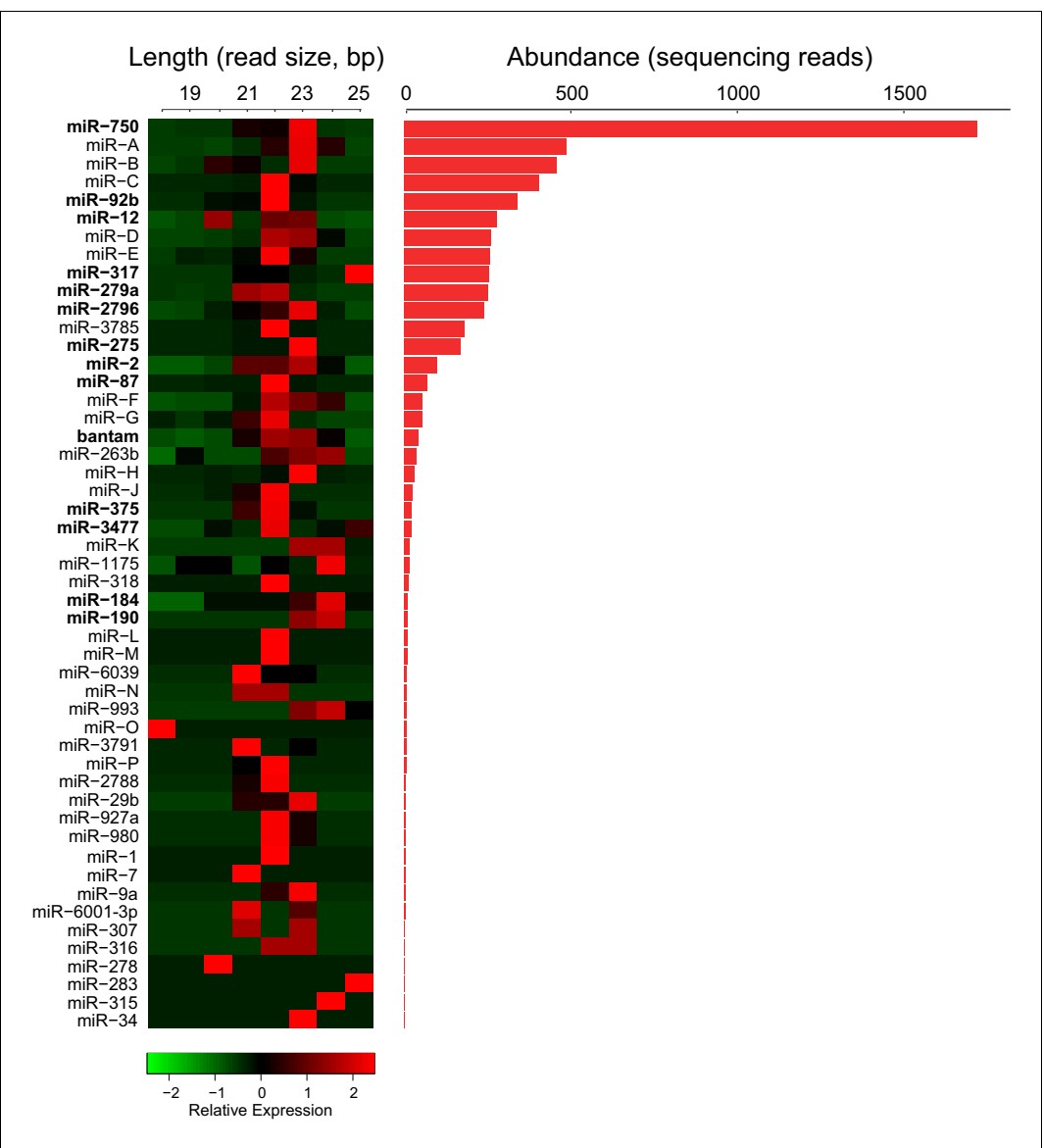

**Figure 2.** Trophallactic fluid of *C. floridanus* contains microRNAs. Left: heatmap showing the length of reads assigned to each *C. floridanus* microRNA (miRNA) found in TF. MiRNAs should exhibit a consistent read size typically between 18 and 22 base pairs. Right: histogram indicating read abundance. miRNA names were assigned through homology to *A. mellifera* where possible. Letters were assigned for novel miRNAs. Bold miRNA names indicate miRNAs whose homologs were also observed in royal and/or worker jelly in *A. mellifera* (***Guo et al., 2013***). Source data in Figure 2—source data 1.

DOI: https://doi.org/10.7554/eLife.20375.006

but the expression of an orthologous miRNA in the Asian tiger shrimp, *Penaeus monodon*, is regulated by immune stress (***Kaewkascholkul et al., 2016***). Notably, 16 of the miRNAs detected in the *C. floridanus* TF are also present in *A. mellifera* worker jelly and/or royal jelly (***Guo et al., 2013***), which are oral secretions fed to larvae to bias them toward a worker or queen fate.

## Trophallactic fluid contains long-chain hydrocarbons

Trophallaxis has long been suggested to contribute to the exchange and homogenization of colony odor (***Crozier and Dix, 1979***). The observation that different ant species have distinct blends of non-volatile, cuticular hydrocarbons (CHCs) – which also display quantitative variation within species – have made these chemicals prime candidates for conveying nestmate recognition cues

(*Sharma et al., 2015*; *Bos et al., 2011*; *Hefetz, 2007*; *Lahav et al., 1999*; *Ozaki et al., 2005*; *van Zweden and d'Ettorre, 2010*). CHC profiles of ants within a colony are similar, while individuals isolated from their home colony often vary in their CHC make-up (*Boulay and Lenoir, 2001*), suggesting that the profiles are constantly unified between nestmates (*Boulay et al., 2000a*). However, it remains unclear if CHCs are exchanged principally by trophallaxis, or through other mechanisms such as allo-grooming and contact with the nest substrate (*Boulay et al., 2000a*; *Soroker et al., 1995*; *Bos et al., 2011*). We therefore investigated whether CHCs are present in TF, and how they relate to those present on the cuticle. To do this, we analyzed by gas chromatography mass spectrometry (GC-MS) the TF and cuticular extracts collected from the same five groups of 20–38 workers.

We identified 61 molecules in TF (*Table 1*), including fatty acids and fatty acid esters, linear alkanes, double bonded hydrocarbons, branched hydrocarbons, and a cholesterol-like molecule. The most abundant TF compounds comprised 27 or more carbons. Cuticular extracts also contained predominantly multiply branched alkanes with 27 or more carbons, corresponding to previous observations of CHCs in this species (*Sharma et al., 2015*; *Endler et al., 2004*) (*Figure 3*). All the highly abundant hydrocarbons in TF (marked in *Figure 3*) were also present on the cuticle of the workers analyzed, supporting the potential for trophallaxis to mediate inter-individual CHC exchange. Moreover, there was a significantly greater (t-test, p<0.0003) similarity across colonies in the hydrocarbon profiles of TF than hydrocarbon profiles of the cuticle (similarity was measured by cross-correlation between GC-MS profiles, *Figure 3D–E*). Altogether, these observations indicate that while CHCs are likely to be exchanged by trophallaxis, additional mechanisms are probably involved in generating the colony-specific bouquet of these compounds.

## Juvenile hormone is exchanged through trophallaxis and can influence larval development

Given the abundance of the expanded family of JHE-like proteins in TF, we asked whether JH itself is also present in this fluid. The primary JH found in Hymenoptera, JH III, is thought to circulate in the hemolymph after being produced by the corpora allata (*Wigglesworth, 1936*). To detect and quantify JH in both TF and hemolymph, we employed a derivatization and purification process prior to GC-MS analysis (*Brent and Vargo, 2003*; *Shu et al., 1997*; *Bergot et al., 1981*). While both sets of measurements were highly variable across samples, we found high levels of JH in TF with concentrations of the same order of magnitude as those found in hemolymph (*Figure 4A*).

Because JH is an important regulator of development, reproduction and behavior (*Nijhout and Wheeler, 1982*), we investigated whether the dose of JH that a larva receives by trophallaxis with a nursing worker might be physiologically relevant. If an average worker has approximately 0.34 μL of TF in her crop at a given time (as measured in our initial experiments; see above), this corresponds to a dose of approximately 31 pg of JH. The analysis of 35 larvae collected midway through development (i.e., third instar out of four worker instars, mean ± SD: 4.0 ± 0.19 mm long and 1.4 ± 0.12 mm wide) revealed that they contained 100–700 pg of JH (*Figure 4B*). Thus, the amount of JH received during an average trophallaxis event amounts to 5–31% of the JH content of a third instar larva. While it is difficult to determine how much of the JH fed to larvae remains in the larval digestive tract, these results indicate that there is potentially sufficient JH in a single trophallaxis-mediated feeding to shift the titer of a recipient larva.

We next determined whether adding exogenous JH to the food of nursing workers could change the growth of the reared larvae. We created groups of 25–30 workers and allowed them to each rear 5–10 larvae to pupation, in the presence of food and sucrose solution that was supplemented with either JH or only a solvent. Larvae reared by JH-supplemented workers grew into larger adults than those reared by solvent-supplemented controls (*Figure 4C*, GLMM, p<9.01e$^{-06}$). Moreover, larvae reared by JH-supplemented caretakers were twice as likely to successfully undergo metamorphosis relative to controls (*Figure 4D*, binomial GLMM, p<7.39e$^{-06}$). These results are consistent with previous studies in other species of ants and bees using methoprene (a non-hydrolyzable JH analog), whose external provision to the colony can lead to larvae developing into larger workers and even queens (*Nijhout and Wheeler, 1982*; *Libbrecht et al., 2013*; *Wheeler and Nijhout, 1981*).

**Table 1.** Table of all components identified by GC-MS in TF of *C. floridanus*.
Molecules marked with black dots (•) were found only in TF and not on the cuticle. Peak ID corresponds to *Figure 3*.

| Rt (min) | MW | Proposed MF | Proposed structure | RI(a) | Peak ID |
|---|---|---|---|---|---|
| 13.35 | 212 | C15H32 | Pentadecane | 1500 | |
| 14.05 | 226 | C16H34 | 5-methylpentadecane | 1541 | |
| 15.05 | 226 | C16H34 | Hexadecane | 1600 | |
| 16.47 | 240 | C17H36 | Heptadecane | 1700 | |
| 17.06 | 254 | C18H38 | 7-methylheptadecane | 1742 | |
| 17.83 | 254 | C18H38 | Octadecane | 1800 | |
| 19.15 | 268 | C19H40 | Nonadecane | 1900 | |
| 19.89 | 295 | C21H44 | *-trimethyloctadecane | 1950 | |
| 19.97 | 256 | C16H32O2 | n-Hexadecanoic acid | 1956 | |
| 20.17 | 282 | C18H34O2 | Hexadecenoic acid, ethyl ester | 1970 | |
| 20.50 | 284 | C18H36O2 | Ethyl palmitate | 1992 | |
| 20.62 | 282 | C20H42 | Eicosane | 2000 | |
| 20.98 | 268 | C18H36O | Octadecanal | 2020 | |
| 22.89 | 280 | C18H32O2 | Linoleic acid | 2125 | |
| 23.03 | 282 | C18H34O2 | Oleic acid | 2137 | • |
| 23.46 | 308 | C20H36O2 | *-Octadecadienoic acid, ethyl ester (possibly Ethyl linoleate) | 2155 | |
| 23.59 | 310 | C20H38O2 | Ethyl oleate | 2165 | • |
| 24.30 | 310 | C22H46 | Docosane | 2200 | |
| 26.27 | 322 | C23H46 | *-tricosene | 2279 | • |
| 26.56 | 324 | C23H48 | Tricosane | 2300 | |
| 29.08 | 338 | C24H50 | Tetracosane | 2400 | |
| 30.76 | 352 | C25H52 | 2-methyltetracosane | 2461 | |
| 31.05 | 350 | C25H50 | *-pentacosene | 2472 | |
| 31.26 | 350 | C25H50 | *-pentacosene | 2480 | |
| 31.80 | 352 | C25H52 | Pentacosane | 2500 | |
| 34.69 | 366 | C26H54 | Hexacosane | 2600 | |
| 36.37 | 380 | C27H56 | 4-methylhexacosane | 2658 | |
| 36.91 | 378 | C27H54 | *-heptacosene | 2675 | |
| 37.70 | 380 | C27H56 | Heptacosane | 2700 | |
| 39.19 | 394 | C28H58 | 5-methylheptacosane | 2755 | |
| 40.63 | 394 | C28H58 | Octacosane | 2800 | |
| 41.36 | 422 | C30H62 | *-trimethylheptacosane | 2835 | |
| 41.88 | 408 | C29H60 | 4-methyloctacosane | 2860 | |
| 42.23 | 406 | C29H58 | *-nonacosene | 2877 | |
| 42.47 | 422 | C30H62 | 2,10-dimethyloctacosane | 2889 | G |
| 42.70 | 408 | C29H60 | Nonacosane | 2900 | H |
| 42.90 | 422 | C30H62 | *-dimethyloctacosane | 2918 | |
| 43.29 | 422 | C30H62 | 9-methylnonacosane | 2938 | |
| 43.36 | 422 | C30H62 | 7-methylnonacosane | 2942 | |
| 43.51 | 422 | C30H62 | 5-methylnonacosane | 2951 | |
| 43.79 | 436 | C31H64 | 7,11-dimethylnonacosane | 2968 | |
| 43.90 | 422 | C30H62 | 3-methylnonacosane | 2976 | E |
| 43.96 | 436 | C31H64 | 5,9-dimethylnonacosane | 2980 | K |

*Table 1 continued on next page*

*Table 1 continued*

| Rt (min) | MW | Proposed MF | Proposed structure | RI(a) | Peak ID |
|---|---|---|---|---|---|
| 44.12 | 450 | C32H66 | *-trimethylnonacosane | 2991 | |
| 44.42 | 436 | C31H64 | 3,7-dimethylnonacosane | 3008 | B |
| 44.74 | 450 | C32H66 | 3,7,11-trimethylnonacosane | 3036 | Q |
| 45.01 | 464 | C33H68 | 3,7,11,15-tetramethylnonacosane | 3056 | |
| 45.05 | 436 | C31H64 | *-methyltriacontane (likely 4-) | 3060 | R |
| 45.30 | 434 | C31H62 | *-hentriacontene | 3074 | |
| 45.45 | 450 | C32H66 | 4,10-dimethyltriacontane | 3088 | S |
| 45.54 | 386 | C27H46O | Cholesterol-like | 3090 | A |
| 45.70 | 436 | C31H64 | Hentriacontane | 3100 | M |
| 46.00 | 464 | C33H68 | *-dimethylhentriacontane (likely 9,13) | 3139 | C |
| 46.56 | 464 | C33H68 | 5,9-dimethylhentriacontane | 3187 | D |
| 46.69 | 492 | C35H72 | *-pentamethyltriacontane (possibly 7,11,15,19,23) | 3199 | J |
| 46.93 | 478 | C34H70 | *-trimethylhentriacontane (likely 3,7,11-) | 3223 | O |
| 47.15 | 492 | C35H72 | 5,9,11,15-tetramethylhentriacontane | 3245 | N |
| 47.59 | 492 | C35H72 | *-trimethyldotriacontane or *-tetramethylhentriacontane | 3289 | P |
| 48.78 | 506 | C36H74 | *-Trimethyltritriacontane (possibly 9,13,17) | 3412 | L |
| 49.67 | 520 | C37H76 | *-trimethyltetratriacontane | 3500 | |
| 50.28 | 534 | C38H78 | *-tetramethyltetratriacontane | 3554 | |

DOI: https://doi.org/10.7554/eLife.20375.011

## Comparative proteomics reveals species-specific growth-regulatory proteins in trophallactic fluid

To expand our survey of TF, we collected this fluid from other species of social insects: a closely related ant (*C. fellah)*, an ant from another sub-family (the fire ant *Solenopsis invicta*) and the honey bee *A. mellifera*. Nano-LC-MS/MS analyses identified 79, 350 and 136 proteins in these three species, respectively (84 were identified in *C. floridanus*). We assigned TF proteins from all four analyzed species to 138 distinct groups of predicted orthologous proteins (see Materials and methods); of these, 72 proteins were found in the TF of only one species (*Figure 5*, *Figure 5—source data 1*).

Only eight ortholog groups contained representatives present in the TF of all four species (*Figure 5B*). Most of these appear to have functions related to digestion, except for apolipophorin, which is involved in lipid/nutrient transport (*Kutty et al., 1996*). For ortholog groups found in the TF of the three ant species, most were also digestion-related with the notable exception of CREG1, a secreted glycoprotein that has been implicated in cell growth control (*Di Bacco and Gill, 2003*) and insect JH response (*Li et al., 2007*; *Barchuk et al., 2007*; *Zhang et al., 2014*). Genus- and species-specific proteins were frequently associated with growth or developmental roles. For example, *A. mellifera* TF contained 12 distinct major royal jelly proteins (MRJP), which are thought to be involved in nutrient storage and developmental fate determination (*Drapeau et al., 2006*; *Kamakura, 2011*), while *S. invicta* TF contained a highly abundant JH-binding protein and a vitellogenin. The two *Camponotus* species shared many orthologous groups, consistent with their close phylogenetic relationship. Five of the seven JHE/Est-6 proteins found by Scaffold in *C. floridanus* TF also had orthologs present in *C. fellah* TF. Additionally, these two species shared a MRJP/Yellow homolog, and an NPC2-related protein, which, in *D. melanogaster*, is involved in sterol binding and ecdysteroid biosynthesis (*Huang et al., 2007*). Finally, 26 ortholog groups contained representatives from multiple species, but not the most closely related ones (*e.g.*, *A. mellifera* and *S. invicta,* or *S. invicta* and only one of the two *Camponotus* species). One-third of these were associated with growth and developmental processes (e.g., three hexamerins, two MRJP/yellow proteins, imaginal disc growth factor 4, vitellogenin-like Vhdl and an additional NPC2). Together these analyses indicate that approximately half of all TF protein ortholog groups appear to be digestion-related, consistent with TF being

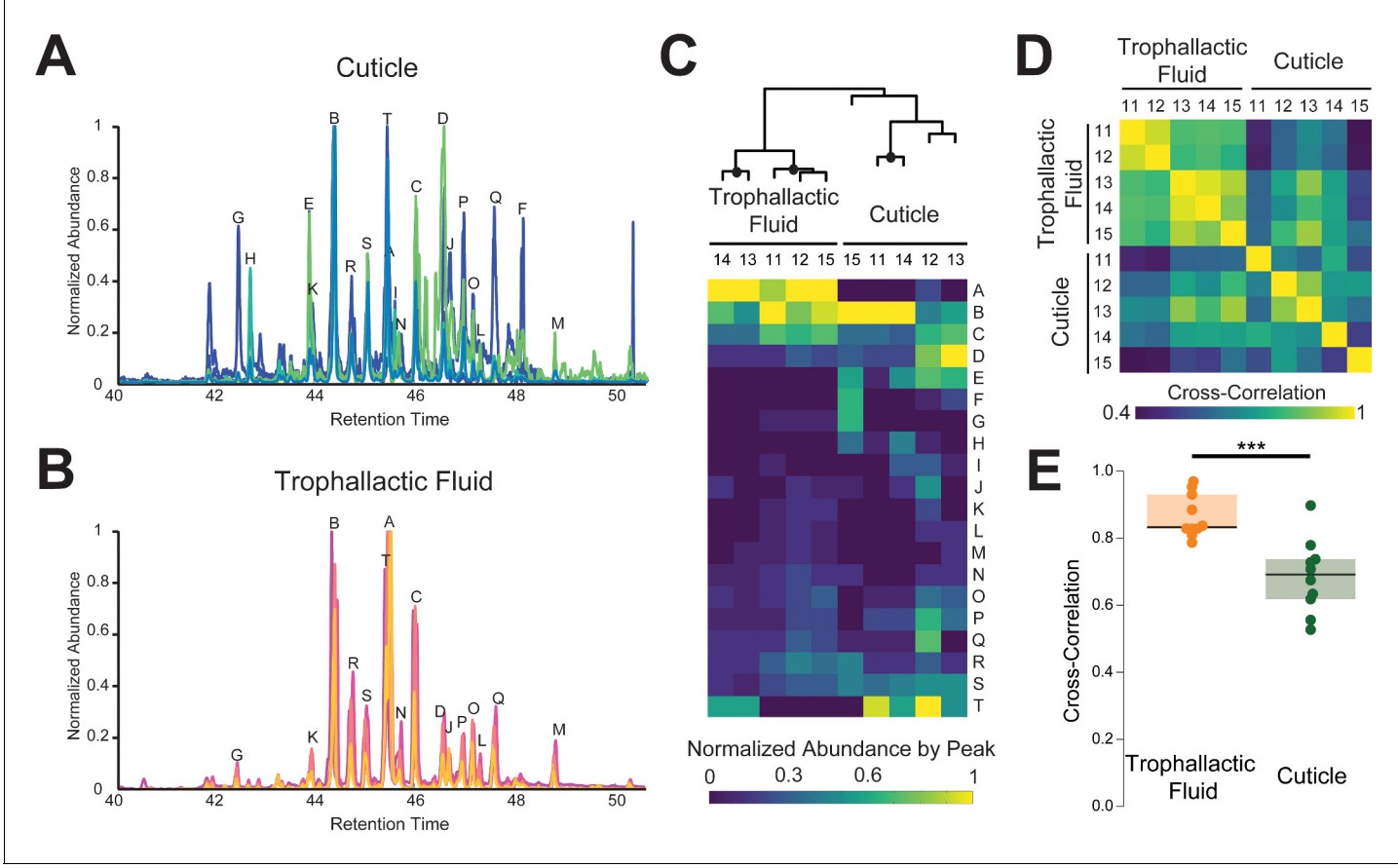

**Figure 3.** Trophallactic fluid of *C. floridanus* contains cuticular hydrocarbons. (**A–B**) Gas chromatography-mass spectrometry profiles in the retention time window for cuticular hydrocarbons (C28–C37), from hexane extracts of whole body (**A**) and from trophallactic fluid (**B**). Samples were extracts from whole body and trophallactic fluid for five groups of 20–38 ants. Each group of ants is from a different colony, C11-C15. Different colonies are shown in distinct colors. Source data in *Figure 3—source data 1*. The abundant component (peak A) found in TF samples but not on the cuticle was a cholesterol-like molecule that insects cannot synthesize but must receive from their diet. Three molecules outside this window were found only in TF and not on the cuticle: *-tricosene, oleic acid, ethyl oleate (*Table 1*). All have been reported to be pheromones in other insect species (*Wang et al., 2011*; *Le Conte et al., 2001*; *Mohammedi et al., 1996*; *Choe et al., 2009*). (**C**) A hierarchically clustered heatmap of the dominant peaks in the range of retention times for long-chain cuticular hydrocarbons. The dendrogram shows approximately unbiased probabilities for 10,000 repetitions. Approximately unbiased bootstrap values > 95% are indicated with black circles. Letters along the right correspond to individual peaks in (**A**) and (**B**). (**D**) Normalized pair-wise cross-correlation values for each TF and body hydrocarbon profile for each of the five colonies. Source data in *Figure 3— source data 2*. (**E**) Normalized pair-wise cross-correlation values between TF hydrocarbon profiles and between body hydrocarbon profiles indicate that the TF hydrocarbon profiles are significantly more similar than are body hydrocarbon profiles. Median values and interquartile ranges are shown. t-test, p<0.0003.

DOI: https://doi.org/10.7554/eLife.20375.007

The following source data and figure supplement are available for figure 3:

**Source data 1.** Peak lists for cuticular and trophallactic fluid long chain hydrocarbons form GC-MS experiments for five different colonies of *C.floridanus*.
DOI: https://doi.org/10.7554/eLife.20375.009
**Source data 2.** Cross-correlation matrix for panels 3D-E.
DOI: https://doi.org/10.7554/eLife.20375.010
**Figure supplement 1.**
DOI: https://doi.org/10.7554/eLife.20375.008

composed of the contents of the foregut. However, many TF proteins have putative roles in growth, nutrient storage, or the metabolism and transport of JH, vitellogenin or ecdysone.

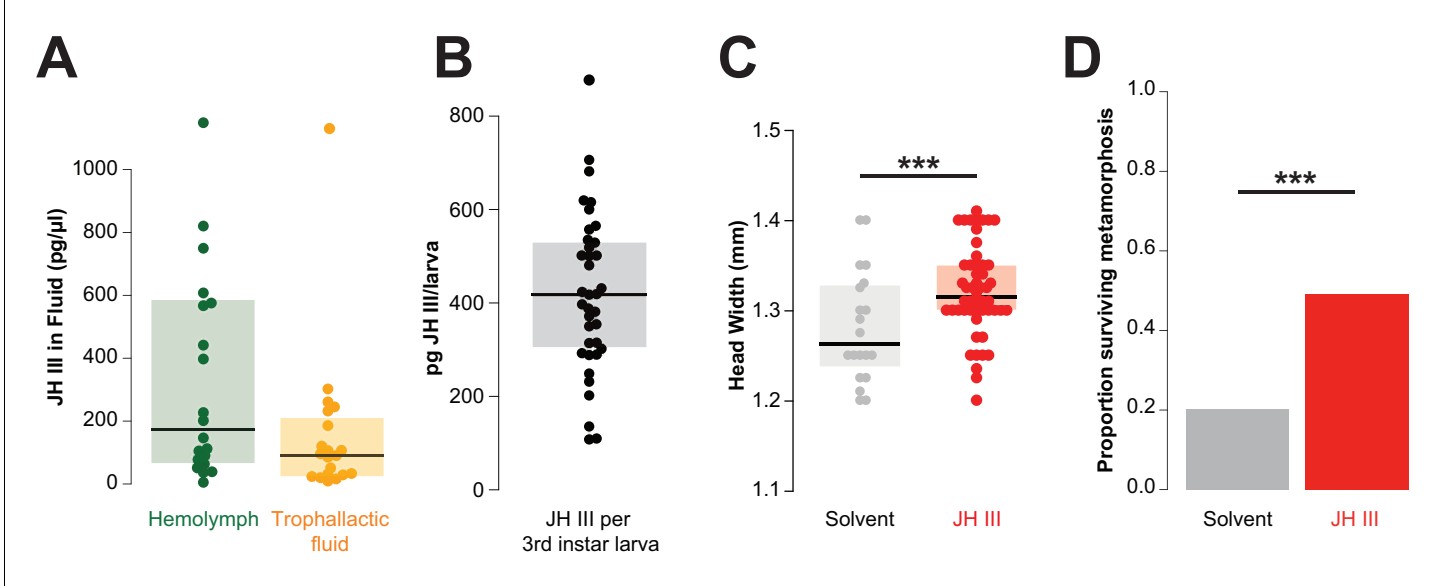

**Figure 4.** Juvenile hormone passed in trophallactic fluid increases larval growth and rate of pupation in *C. floridanus*. (A) JH titer in trophallactic fluid and hemolymph (n = 20; each replicate is a group of 30 workers). Source data in *Figure 4—source data 1*. (B) JH content of third instar larvae. Source data in *Figure 4—source data 2*. (C) Head width of pupae raised by workers who were fed food supplemented with JH III or solvent. General linear mixed model (GLMM) testing effect of JH on head width with colony, replicate and experiment as random factors, ***$p < 9.01e^{-06}$. Source data in *Figure 4—source data 3*. (D) Proportion of larvae that have undergone metamorphosis when workers were fed food supplemented with JH III or solvent only. Binomial GLMM testing effect of JH on survival past metamorphosis with colony and experiment as random factors, ***$p < 7.39e^{-06}$. Median values and interquartile ranges are shown in panels (A–C). Panels (C) and (D) are data from three separate experiments where effects in each were individually significant to $p<0.05$. Source data in *Figure 4—source data 4*.

DOI: https://doi.org/10.7554/eLife.20375.012

The following source data is available for figure 4:

**Source data 1.** Hemolymph and trophallactic fluid Juvenile hormone titers for 20 pooled samples of each fluid.
DOI: https://doi.org/10.7554/eLife.20375.013
**Source data 2.** Juvenile hormone titers for 37 individual third instar larvae.
DOI: https://doi.org/10.7554/eLife.20375.014
**Source data 3.** Head-width measurements for panel 4C.
DOI: https://doi.org/10.7554/eLife.20375.015
**Source data 4.** Metamorphosis or death counts for panel 4D.
DOI: https://doi.org/10.7554/eLife.20375.016

## Discussion

We have characterized the fluid that is orally exchanged during trophallaxis, a distinctive behavior of eusocial insects generally considered as a means of food sharing. Our results reveal that the transmitted liquid contains much more than food and digestive enzymes, and includes non-proteinaceous and proteinaceous molecules implicated in chemical discrimination of nestmates, growth and development, and behavioral maturation. These findings suggest that trophallaxis underlies a private communication mechanism that can have multiple phenotypic consequences. More generally, our observations open the possibility that exchange of oral fluids (e.g., saliva) in other animals might also serve functions not previously suspected (*Humphrey and Williamson, 2001*; *Ribeiro, 1995*).

In ants, trophallaxis has long been thought to be a mode of transfer for the long-chain hydrocarbons that underlie nestmate recognition (*Boulay et al., 2000a Dahbi et al., 1999*; *Lahav et al., 1999*; *van Zweden and d'Ettorre, 2010*; *Soroker et al., 1995*; *Lalzar et al., 2010*). However, previous work has analyzed only the passage of radio-labeled hydrocarbons between individuals (*Boulay et al., 2000a*; *Dahbi et al., 1999*; *Soroker et al., 1995*) and between individuals and substrates (*Bos et al., 2011*). Without explicitly sampling the contents of the crop, it is not possible to differentiate between components passed by trophallaxis or by physical contact. Our study is, to our

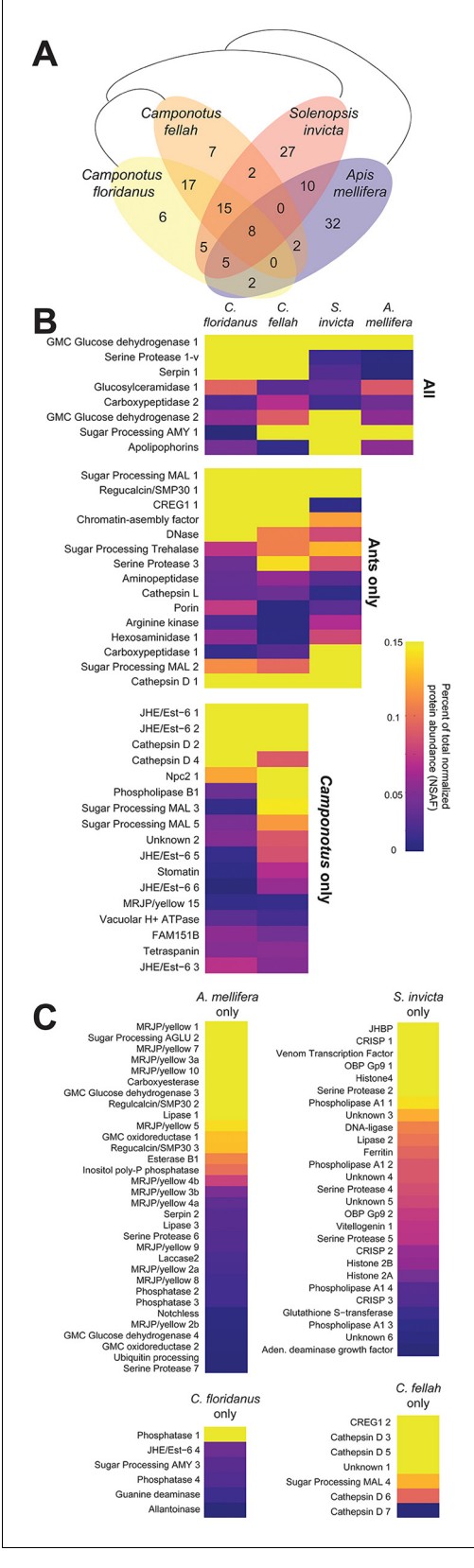

**Figure 5.** Proteins in trophallactic fluid across social insect species. (A) Venn diagram indicating the number of species-specific and orthologous proteins detected *Figure 5 continued on next page*

knowledge, the first to demonstrate that TF contains endogenous long-chain hydrocarbons. Interestingly, we found differences in the hydrocarbon profiles of the TF and the cuticle of the same ants, suggesting the existence of other processes regulating the relative proportion of CHCs and/or additional sources of these compounds. Further work manipulating and comparing hydrocarbon profiles of TF and cuticle of multiple individual ants within the same colony is necessary to understand the impact of the observed variation in TF and cuticular hydrocarbons.

Consistent with the view that the gut is one of the first lines of defense in the body's interface with the outside world (*Lemaitre and Hoffmann, 2007*; *Söderhäll and Cerenius, 1998*), the four TF proteomes analyzed in this study include many potential defense-related proteins. Homologs of the Cathepsin D family of proteins, which have been implicated in growth, defense and digestion (*Saikhedkar et al., 2015*; *Ahn and Zhu-Salzman, 2009*), were present in the TF of both *Camponotus* species. Prophenoloxidase 2, an enzyme responsible for the melanization involved in the insect immune response (*Söderhäll and Cerenius, 1998*), was found in the TF of *A. mellifera* and *S. invicta*. All four species contained several members of the serine protease and serpin families, which have well-documented roles in the prophenoloxidase cascade and more general immune responses in *D. melanogaster* and other animals (*Lemaitre and Hoffmann, 2007*; *Lucas et al., 2009*). The TF of two species had orthologs of the recognition lectin GNBP3 which is also involved in the prophenoloxidase cascade. Finally, a handful chromatin-related proteins (e.g., histones, CAF1) were found in *S. invicta*, and to a lesser extent in the other two ants. Their presence may simply reflect a few cells being sloughed from the lumen of the foregut, or could be indicative of a defense process termed ETosis, whereby chromatin is released from the nuclei of inflammatory cells to form extracellular traps that kill pathogenic microbes (*Robb et al., 2014*; *Brinkmann et al., 2004*).

The presence of development- and growth-related components in the TF of diverse social insects suggests that this fluid may play a role in directing larval development. Previous work has shown that the developmental fate of larvae and the process of caste determination in various ant and bee species can be influenced by treating colony members with JH or JH analogs (*Wheeler, 1986*; *Wheeler and Nijhout, 1981*; *Cnaani et al., 1997*; *Cnaani et al., 2000*;

*Figure 5 continued*

in TF from the indicated species, whose phylogenetic relationships are shown with black lines. (B) Heat maps showing the percentage of total molecular-weight normalized spectra in TF samples assigned to the proteins in each given species, averaged over all in-colony samples for that species. Samples sizes: *C. floridanus* (n = 15), *C. fellah* (n = 6), *S. invicta* (n = 3), *A. mellifera* (n = 6). (C) Species-specific TF proteins. The 26 TF ortholog groups found in two or three species, but not the most closely related ones (e.g., *A. mellifera* and *S. invicta,* or *S. invicta* and only one of the two *Camponotus* species) are indicated in *Figure 5— figure supplement 1*.

DOI: https://doi.org/10.7554/eLife.20375.017

The following source data and figure supplement are available for figure 5:

**Source data 1.** A table of all orthologous proteins and their predicted functions, identifiers, known *D. melanogaster* orthologs, presence of annotated secretion signals and average NSAF values when present in TF.
DOI: https://doi.org/10.7554/eLife.20375.018

**Figure supplement 1.** Heat map showing the percentage of total molecular-weight normalized spectra in trophallactic fluid samples assigned to each protein in three ant species and the European honey bee, averaged over all in-colony samples for that species.
DOI: https://doi.org/10.7554/eLife.20375.019

*Rajakumar et al., 2012*). Moreover, in some species, workers play an important role in regulating caste determination (*Kamakura, 2011*; *Linksvayer et al., 2011*; *Villalta et al., 2016*; *Mutti et al., 2011*) but it is unknown how workers might influence the JH titers of larvae. Our finding that TF contains JH raises the possibility of trophallaxis as a direct means by which larval hormone levels and developmental trajectories can be manipulated. This type of mechanism has some precedent: honey bee workers bias larval development toward queens by feeding them royal jelly, an effect that might be mediated by MRJPs (*Kamakura, 2011*; *Buttstedt et al., 2016*; *Kucharski et al., 2015*; *Kamakura, 2016*). Across the TF of four species of social insect, we have found diverse molecules intimately involved in insect growth regulation: JH, JHE, JH-binding protein, vitellogenin, hexamerin, apolipophorin, and MRJPs. Several of the proteins and microRNAs identified in ant and bee TF are also found in royal and worker jelly (*Guo et al., 2013*; *Zhang et al., 2014*). Notably, there are also some similarities between proteins in social insect TF and mammalian milk (*Zhang et al., 2015*), such as the cell growth regulator CREG1. The between-genera variation in the most abundant growth-related proteins (e.g., MRJPs in honey bee, JH-binding protein and hexamerins in fire ant, and JHEs in *Camponotus*) indicates that there might be multiple evolutionary origins and/or rapid divergence in trophallaxis-based signals potentially influencing larval growth.

The finding that TF contains many proteins, miRNAs, CHCs and JH raises the question of how they come to be present in this fluid. Many non-food-derived components in TF are likely to be either directly deposited into the alimentary canal: over two-thirds of the proteins present in TF had a predicted signal peptide suggesting that they are probably secreted by the cells lining the foregut or glands connected to the alimentary canal (e.g., postpharyngeal, labial, mandibular, salivary glands). Additionally, TF molecules are likely to be acquired by transfer among nestmates through licking, grooming, and trophallaxis. For example, CHCs are produced by oenocytes, transmitted to the cuticle, ingested, and sequestered in the postpharyngeal gland (*Bos et al., 2011*; *van Zweden and d'Ettorre, 2010*; *Soroker and Hefetz, 2000*). Interestingly, JH is synthesized in the corpora allata, just caudal to the brain (*Goodman and Granger, 2009*). Although the mechanisms of JH uptake by tissues are still incompletely understood (*Parra-Peralbo and Culi, 2011*; *Rodríguez-Vázquez et al., 2015*; *Engelmann and Mala, 2000*; *Suzuki et al., 2011*), the TF proteins apolipophorin, hexamerin, JH-binding protein and vitellogenin have all been implicated in JH binding and transport between hemolymph and tissues (*Goodman and Granger, 2009*; *Rodríguez-Vázquez et al., 2015*; *Engelmann and Mala, 2000*; *Suzuki et al., 2011*; *Amsalem et al., 2014*), and these factors may be responsible for transporting JH into the foregut. Unfortunately, even in *Drosophila* mechanisms of JH uptake by tissues are still unclear (*Rodríguez-Vázquez et al., 2015*; *Engelmann and Mala, 2000*; *Suzuki et al., 2011*; *Parra-Peralbo and Culi, 2011*). Extracellular miRNAs are secreted and transported through a variety of pathways, but the functional relevance of such molecules is still controversial (*Sarkies and Miska, 2013*; *Turchinovich et al., 2016*; *Masood et al., 2016*; *Søvik et al., 2015*; *Rayner and Hennessy, 2013*).

While the simultaneous presence of JH and a set of putative JH degrading enzymes (the JHE/Est-6 family) in *C. floridanus* TF may appear surprising, there are at least two possible explanations. First, this might reflect a regulatory mechanism for JH levels at the individual and colony levels. Many

biological systems use such negative feedback mechanisms to buffer signals and enable rapid responses to environmental change (*Alon, 2007*). Alternatively, these enzymes may have evolved a different function along with their novel localization in TF. In other insects JHE-related enzymes are typically expressed in the fat body and circulate in the hemolymph (*Hammock et al., 1975*; *Ward et al., 1992*; *Campbell et al., 1998*)., The expansion of JHE-like proteins in *C. floridanus* was accompanied by a high specificity in their localization, with two (E2AM67, E2AM68) being present exclusively in the hemolymph, and four (E2ANU0, E2AI90, E2AJM0, E2AJL9) exclusively in TF. This coexistence of JH and JHEs reveals a striking parallel with the constituents of a different socially exchanged fluid in *D. melanogaster*. *Drosophila* Est-6 is highly abundant in seminal fluid, together with its presumed substrate, the male-specific pheromone 11-*cis*-vaccenyl acetate (cVA). cVA has multiple roles in influencing sexual and aggressive interactions and its transfer to females during copulation potently diminishes her attractiveness to future potential mates (*Poiani, 2006*; *Chertemps et al., 2012*; *Costa, 1989*).

Given that larvae are fed JH and food, and both are necessary for development, a future goal will be to dissect the relative contribution and potential synergy of these components. Furthermore, JH has many other functions in social insects, including behavioral modulation, longevity, fecundity and immunity (*Flatt et al., 2005*; *Jindra et al., 2013*; *Dolezal et al., 2012*; *Wang et al., 2012*); the relevance of trophallaxis-mediated JH exchange between adults remains to be explored. Considering that most, if not all, individuals in a colony must share food through trophallaxis, it will also be of interest to understand if and how individual-, or caste-specific TF-based information signals emerge. Recent work on trophallaxis networks (*Greenwald et al., 2015*) and interaction networks (*Mersch et al., 2013*), indicate that ants preferentially interact with others of similar behavioral type (e.g., nurses with nurses, and foragers with foragers). This raises the possibility that different pools of TF with different qualities exist. A key challenge will be to develop specific genetic and/or pharmacological tools to test the biological relevance of molecules transmitted by trophallaxis.

## Materials and methods

### Insect source and rearing

*Camponotus floridanus* workers came from 16 colonies established in the laboratory from approximately 1-year-old founding queens and associated workers collected from the Florida Keys in 2006, 2011 and 2012. Ants were provided once a week with fresh sugar water, an artificial diet of honey or maple syrup, eggs, agar, canned tuna and a few *D. melanogaster*. Maple syrup was substituted for honey and no *Drosophila* were provided in development and proteomic experiments to avoid contamination with other insect proteins. Colonies were maintained at 25°C with 60% relative humidity and a 12 hr light:12 hr dark cycle. *Camponotus fellah* colonies were established from queens collected after a mating flight in March 2007 in Tel Aviv, Israel (Colonies #5, 28, 33). The ant colonies were maintained at 32°C with 60% relative humidity and a 12 hr light:12 hr dark cycle. Fire ant workers (*Solenopsis invicta*) were collected from three different colonies, two polygyne, one monogyne, maintained at 32°C with 60% relative humidity and a 12 hr light:12 hr dark cycle. Honeybee workers (*Apis mellifera*, Carnica and Buckfast) were collected from six different hives maintained with standard beekeeping practices.

### Fluid collection

'Voluntary' TF was collected from individuals who had been starved and socially isolated for 1–3 weeks, then fed 25% sucrose solution, and promptly re-introduced to other separately isolated nestmates. Ants were monitored closely for trophallaxis events; when one had begun, a pulled glass pipette was brought between the mouthparts of the 2–4 individuals and fluid was collected. Typically this stopped the trophallaxis event, but on rare occasions it was possible to acquire up to submicroliter volumes of TF.

For 'forced' collection, ants were anesthetized by $CO_2$ (on a $CO_2$ pad; Flypad, FlyStuff, San Diego, CA) and promptly flipped ventral-side up. The abdomen of each ant was lightly squeezed with wide insect forceps to prompt the regurgitation of fluids from the social stomach. Ants that underwent anesthesia and light squeezing yielded on average 0.34 µL of fluid and recovered in approximately 5 min. TF was collected with graduated borosilicate glass pipettes, and transferred

immediately to either buffer (100 mM $NaH_2PO_4$, 1 mM EDTA, 1 mM DTT, pH 7.4) for proteomic experiments, to RNase-free water for RNA analyses, or to pure EtOH for GC-MS measurements. TF was collected from *C. fellah* and *S. invicta* in the same manner as from *C. floridanus*; for *S. invicta* 100–300 ants were used due to their smaller body size and crop volume.

Honey bee TF was collected from bees that were first cold-anesthetized and then transferred to a $CO_2$ pad to ensure continual anesthesia during collection of TF as described above. Honey bees yielded higher volumes of TF than did ants (0.94 µL TF per honey bee, SD = 0.54 relative to 0.34 µL TF per *C. floridanus* ant, SD = 0.27).

Hemolymph was collected from $CO_2$-anesthetized ants by puncturing the junction between the foreleg and the distal edge of the thoracic plate with a pulled glass pipette. This position was chosen over the abdomen in order to ensure that hemolymph was collected and not the contents of the crop. Approximately 0.1 µL was taken from each ant.

Midgut samples were collected by first anesthetizing an ant, immobilizing it in warm wax ventral-side up, covering the preparation in 1x PBS, and opening the abdomen with dissection forceps and iris scissors. The midgut was punctured by a sharp glass pipette and its contents collected. Because the pipette also contacts the surrounding fluid, some hemolymph contamination was unavoidable.

## Proteomic analyses

### Gel separation and protein digestion

Protein samples were loaded on a 12% mini polyacrylamide gel and migrated about 2.5 cm. After Coomassie staining, regions of gel lanes containing visible bands (generally > 18 kDa) were excised into 2–4 pieces, depending on the gel pattern of considered experiment. Gel pieces were digested with sequencing-grade trypsin (Promega, Switzerland) as described (*Shevchenko et al., 2006*). Extracted tryptic peptides were dried and resuspended in 0.1% formic acid, 2% (v/v) acetonitrile for mass spectrometry analyses.

### Proteomic mass spectrometry analyses

Tryptic peptide mixtures were injected on a Dionex RSLC 3000 nanoHPLC system (Dionex, Sunnyvale, CA) interfaced via a nanospray source to a high resolution mass spectrometer based on Orbitrap technology (Thermo Fisher, Bremen, Germany): LTQ-Orbitrap XL ('voluntary' and hemolymph samples), LTQ-Orbitrap Velos (*A. mellifera* TF samples) or QExactive Plus ('voluntary' and all other samples). Peptides were loaded onto a trapping microcolumn Acclaim PepMap100 C18 (20 mm x 100 µm ID, 5 µm, Dionex) before separation on a C18 reversed-phase analytical nanocolumn at a flowrate of 0.3 µL/min. Q-Exactive Plus instrument was interfaced with an Easy Spray C18 PepMap nanocolumn (25 or 50 cm x 75 µm ID, 2 µm, 100 Å, Dionex) using a 35-min gradient from 4% to 76% acetonitrile in 0.1% formic acid for peptide separation (total time: 65 min). Full MS surveys were performed at a resolution of 70,000 scans. In data-dependent acquisition controlled by Xcalibur software (Thermo Fisher), the 10 most intense multiply charged precursor ions detected in the full MS survey scan were selected for higher energy collision-induced dissociation (HCD, normalized collision energy NCE = 27%) and analyzed in the orbitrap at 17,500 resolution. The window for precursor isolation was of 1.5 *m/z* units around the precursor and selected fragments were excluded for 60 s from further analysis.

The LTQ-Orbitrap Velos mass spectrometer was interfaced with a reversed-phase C18 Acclaim Pepmap nanocolumn (75 µm ID x 25 cm, 2.0 µm, 100 Å, Dionex) using a 65 min gradient from 5% to 72% acetonitrile in 0.1% formic acid for peptide separation (total time: 95 min). Full MS surveys were performed at a resolution of 60,000 scans. In data-dependent acquisition controlled by Xcalibur software, the 20 most intense multiply charged precursor ions detected in the full MS survey scan were selected for CID fragmentation (NCE = 35%) in the LTQ linear trap with an isolation window of 3.0 *m/z* and then dynamically excluded from further selection for 120 s.

LTQ-Orbitrap XL instrument was interfaced with a reversed-phase C18 Acclaim Pepmap (75 µm ID x 25 cm, 2.0 µm, 100 Å, Dionex) or Nikkyo (75 µm ID x 15 cm, 3.0 µm, 120 Å, Nikkyo Technos, Tokyo, Japan) nanocolumn using a 90 min gradient from 4% to 76% acetonitrile in 0.1% formic acid for peptide separation (total time: 125 min). Full MS surveys were performed at a resolution of 60,000 scans. In data-dependent acquisition controlled by Xcalibur software, the 10 most intense multiply charged precursor ions detected in the full MS survey scan were selected for CID

fragmentation (NCE = 35%) in the LTQ linear trap with an isolation window of 4.0 $m/z$ and then dynamically excluded from further selection for 60 s.

## Proteomic data analysis

MS data were analyzed primarily using Mascot 2.5 (RRID:SCR_014322, Matrix Science, London, UK) set up to search either the UniProt (RRID:SCR_002380, www.uniprot.org) or NCBI (RRID:SCR_003496, www.ncbi.nlm.nih.gov) database restricted to *C. floridanus* (UniProt, August 2014 version: 14,801 sequences; NCBI, July 2015 version: 34,390 sequences), *S. invicta* (UniProt, August 2014 version: 14,374 sequences; NCBI, January 2015 version: 21,118 sequences) or *A. mellifera* (UniProt, September 2015 version: 15,323 sequences; NCBI, February 2016 version: 21,772 sequences) taxonomy. For Mascot search of *C. fellah* samples, we used the database provided by the *C. fellah* transcriptome (27,062 sequences) and the UniProt *C. floridanus* reference proteome (January 2016 version, 14,287 sequences). Trypsin (cleavage at K, R) was used as the enzyme definition, allowing two missed cleavages. Mascot was searched with a parent ion tolerance of 10 ppm and a fragment ion mass tolerance of 0.50 Da (LTQ-Orbitrap Velos / LTQ-Orbitrap XL) or 0.02 Da (QExactive Plus). The iodoacetamide derivative of cysteine was specified in Mascot as a fixed modification. N-terminal acetylation of protein, deamidation of asparagine and glutamine, and oxidation of methionine were specified as variable modifications.

Scaffold software (version 4.4, RRID:SCR_014345, Proteome Software Inc., Portland, OR) was used to validate MS/MS based peptide and protein identifications, and to perform dataset alignment. Peptide identifications were accepted if they could be established at greater than 90.0% probability as specified by the Peptide Prophet algorithm (*Keller et al., 2002*) with Scaffold delta-mass correction. Protein identifications were accepted if they could be established at greater than 95.0% probability and contained at least two identified peptides. Protein probabilities were assigned by the Protein Prophet algorithm (*Nesvizhskii et al., 2003*). Proteins that contained similar peptides and could not be differentiated based on MS/MS analysis alone were grouped to satisfy the principles of parsimony and in most cases these proteins appear identical in annotations.

Quantitative spectral counting was performed using the normalized spectral abundance factor (NSAF), a measure that takes into account the number of spectra matched to each protein, the length of each protein and the total number of proteins in the input database (*Zybailov et al., 2006*). To compare relative abundance across samples with notably different protein abundance, we divided each protein's NSAF value by the total sum of NSAF values present in the sample.

Across forced TF samples, ~10% of detected spectra were identified as peptides by Mascot and Scaffold, while in hemolymph samples ~16% of spectra were identified. Of those identified protein spectra in TF, about 2/3 mapped to the *C. floridanus* proteome, 1/3 to typical proteomic experiment contaminants (e.g., human keratin, trypsin added for digestion of proteins), < 5% to elements from the ants' diet (egg proteins, *D.melanogaster*), and < 0.5% to microbial components including endosymbiont *Blochmannia floridanus.* Of the identified hemolymph protein spectra, 94% mapped to the *C. floridanus* proteome and the remaining 6% to keratins, trypsins, etc. We used PEAKS proteomic software (version 7.5, Bioinformatics Solutions Inc., Waterloo, Canada) for a more in-depth analysis of a typical TF sample, making less stringent de novo protein predictions allowing substitutions and post-translational modifications, and we could identify approximately 50% of spectra as peptides.

All proteomics data are available through ProteomeXchange at PXD004825.

Secretion signals were predicted using SignalP 4.1 (*Petersen et al., 2011*).

## Social isolation

Groups of 25–30 *C. floridanus* worker ants were taken from colonies C3, C6, C7, C8, C9 and C10 (queens all collected in the Florida Keys December 2012). Upon collection, TF was collected under $CO_2$ anesthesia. Immediately after TF collection ants were collectively isolated from their home colony and queen in fluon-coated plastic boxes with shelter, insect-free food (maple syrup, chicken eggs, tuna and agar), and water. Ants were kept in these group-isolated queenless conditions for 14 days, after which, their TF was collected again.

## Small RNA analysis

### Total RNA isolation and quantification

Total RNA was isolated (*Sapetschnig et al., 2015*) from approximately 100 µL of TF from *C. floridanus* ants by the Trizol-LS method according to the manufacturer's protocols (Life Technologies, Inc., Grand Island, NY). A Qubit RNA HS Assay Kit (high sensitivity, 5 to 100 ng) was used with a Qubit 3.0 fluorometer according to the manufacturer's protocols (Life Technologies, Carlsbad); a sample volume of 1 µL was added to 199 µL of a Qubit working solution. The RNA concentration was 17.7 ng/µL.

### Library preparation

We used 5 µL of RNA (approximately 88.5 ng) to generate a small RNA sequencing library using reagents and methods provided with TruSeq Small RNA Sample Prep Kit (Illumina, San Diego). Briefly, T4 RNA ligase was used to ligate RA5 and RA3 RNA oligonucleotides to 5' and 3' ends of RNA, respectively. Adapter-ligated RNA was reverse-transcribed using a RTP primer and the resulting cDNA was amplified in an 11-cycle PCR that used RP1 and indexed RP1 primers. We size-selected cDNA libraries using 6% TBE PAGE gels (Life Technologies, Carlsbad, CA) and ethidium bromide staining. Quality of the generated small RNA sequencing library was confirmed using electropherograms from a 2200 TapeStation System (Agilent Technologies, Santa Clara, CA). Desired sizes of cDNA bands were cut from the gel (between 147 and 157 nt), the gel matrix broken by centrifugation through gel breaker tubes (IST Engineering Inc., Milpitas, CA), and cDNA eluted with 400 µL of 0.3M Na-Chloride. Further purification of cDNA was by centrifugation through Spin-X 0.22 µm cellulose acetate filter columns (Corning Costar, Corning, NY) followed by ethanol precipitation. Libraries were sequenced on a HiSeq 2000 Sequencer (Illumina). Small RNA sequencing data are available through SRA database at SRP082161.

### Pre-processing of RNA libraries

Raw sequenced reads from small RNA libraries were submitted to quality filtering and adaptor trimming using cutadapt (version 1.8.1, RRID:SCR_011841, https://pypi.python.org/pypi/cutadapt/1.8.1). Small RNA reads with Phred quality below 20, and fewer than 18 nucleotides after trimming of adaptors, were discarded.

### MicroRNA prediction using miRDeep2

Remaining sequences from small RNA libraries were used with *A. melifera* microRNA database (miRbase version 21) and *C. floridanus* genome (NCBI version 1.0) to predict microRNAs precursors using miRDeep2 (RRID:SCR_012960) with default parameters and GFF (General Feature Format) file extracted by Perl scripts.

### Small RNA reads mapped to *C. floridanus* and *D. melanogaster* genomes

Approximately half the reads that mapped to the *C. floridanus* genome also mapped to *D. melanogaster*, a component of the ants' laboratory diet. While some of these RNAs are likely to be endogenous *C. floridanus* RNAs, we eliminated all reads identical between *C. floridanus* and *D. melanogaster*. Remaining sequences from small RNA libraries were mapped to reference sequences from *C. floridanus* and *D. melanogaster* genomes using Bowtie (version 1.1.1, RRID:SCR_005476, one mismatch allowed). The *C. floridanus* genome (version 1.0) was downloaded from NCBI. The *D. melanogaster* genome (version v5.44) was downloaded from flybase.org. Remaining sequenced reads that did not map to the *C. floridanus* or *D. melanogaster* genome were assembled into contigs with VelvetOptimiser (version 2.2.5; http://bioinformatics.net.au/software.velvetoptimiser.shtml), and BLASTed against non-redundant NCBI bacterial and viral databases to assess their source organism. Hits with an E-value smaller than 1e−5 for nucleotide comparison were considered significant.

### Automatic annotation, penalization, size distribution and gene expression

To perform automatic annotation, we used BedTools (version v2.17.0, RRID:SCR_006646) to compare genomic coordinates from mapped reads against predicted microRNAs, mRNA, tRNA and ncRNA (represented by lncRNA). Reads mapping to both the *C. floridanus* and *D. melanogaster*

genomes were deemed ambiguous and were eliminated. The remaining reads were normalized by reads per million (RPM). The gene expression was measured and normalized by RPM and plotted in heatmap and barplot graphs.

## Gas chromatography mass spectrometry and related sample preparation

Hydrocarbon analysis was performed on trophallactic fluid from five groups of 20–38 ants, each collected from one of five different colonies (C11-C15). TF samples were placed directly into 3:1 hexane:methanol. Immediately after TF collection, body surface CHCs were collected by placing anesthetized ants in hexane for 1 min before removal with cleaned forceps. Methanol was added to the cuticular-extraction hexane (maintaining the 3:1 proportion of the TF samples). The TF and body samples were vortexed for 30 s and centrifuged for 7 min. Hexane fractions were collected using a thrice-washed Hamilton syringe. Samples were kept at −20°C until further analysis.

A Trace 1300 GC chromatograph interfaced with a TSQ 8000 Evo Triple Quadrupole Mass Spectrometer (Thermo Scientific, Bremen, Germany) was used for the study. Hydrocarbons were separated on a 30 m x 0.25 mm I.D. (0.25 mm film thickness) Zebron ZB-5 ms capillary column (Phenomenex, Torrance, CA) using the following program: initial temperature 70°C held for 1 min, ramped to 210°C at 8 °C/min, ramped to 250°C at 2 °C/min, ramped to 300°C at 8 °C/min and held for 5 min. Helium was used as carrier gas at a constant flow of 1 mL/min. Injections of 1 µL of ants' TF or body extracts were made using splitless mode. The injection port and transfer line temperature were kept at 250°C, and the ion source temperature set at 200°C. Ionization was done by electron-impact (EI, 70 eV) and acquisition performed in full scan mode in the mass range 50–550 $m/z$ (scan time 0.2 s). Identification of hydrocarbons was done using XCalibur and NIST 14 library. The TIC MS was integrated and the area reported as a function of Retention time (Rt, min) for each peak.

## Identification of trophallactic fluid hydrocarbons

Characterization of branched alkanes by GC-MS remains a challenge due to the similarity of their electron impact (EI) mass spectra and the paucity of corresponding spectra listed in EI mass spectra databases. A typical GC-MS chromatogram (*Figure 3—figure supplement 1A*) reveals the complexity of the TF sample. The workflow described here was systematically used to characterize the linear and branched hydrocarbons present in TF samples summarized in *Table 1*. The parent ion was first determined for each peak after background subtraction. Ambiguities remained in some cases due to the low intensity or absence of the molecular ion.

Linear alkanes present in TF samples were localized using a standard mixture of C8-C40 alkanes. Then RI values were deduced for all compounds present in the samples based on their retention times (*Figure 3—figure supplement 1G*, red).

To determine the number of methyl branches for alkanes, we examined the distribution of fragment ions in the spectrum by fitting their intensities with an exponential decay function and specifically looking for the fragment ions that do not fit to the calculated exponential decay function. From the experimental mass spectrum (*Figure 3—figure supplement 1B–C*), the intensity of all fragment ions was extracted and fitted with the function (*Figure 3—figure supplement 1D–E*). *Figure 3—figure supplement 1* clearly shows two different resulting EI mass spectra profiles: on the left a linear alkane corresponding to n-hexacosane (Rt = 34.69 min, MF $C_{26}H_{54}$); on the right, a monomethyl-branched structure most likely corresponding to 9-methylnonacosane (Rt = 43.29 min, MF $C_{30}H_{62}$) with two enhanced fragment ions at m/z 141 and 309 emerging from the curve.

Extracting and fitting the fragment ion profiles first helped to discriminate between linear and branched hydrocarbons, but also to estimate the number of methyl branches. To confirm for each compound the number of carbons and branches obtained, we used Kovats retention index (RI) values in the NIST Chemistry WebBook (*Linstrom and Mallard, 2000*). Based on the RI values for similar GC stationary phase from C15 to C38 hydrocarbons, six different curves of RI vs. number of carbons were constructed from linear to pentamethylated alkanes (*Figure 3—figure supplement 1F*). To construct the curves, average RI values were taken of all hydrocarbons available in the database, with a given number of carbons and a given number of methyl branches. For example, the RI value of 2409 obtained for $C_{25}$ and two branches is the average of 3,(7/9/11/13)-dimethyltricosane,

3,(3/5)-dimethyltricosane and 5,(9/11)-dimethyltricosane values listed in the NIST Chemistry Web-Book for the same stationary phase. Those curves were used to check for every compound that, from the measured RI value and the number of carbons found, the number of ramifications found fit properly with the curves.

Once the number of branches and the parent ion mass were known, the position of the different branches could be deduced from the fragment ion values. When ambiguity remained on the position of the branch or double bond, this is indicated with an asterisk in *Table 1*.

The plot of experimental retention times for each compound as a function of the RI index closely fitted the plot using RI values given by NIST for the identified compounds (*Figure 3—figure supplement 1G*), bringing additional confidence to the identifications.

*Table 1* summarizes the proposed structures for hydrocarbons and other compounds detected in TF samples.

## JH quantification by GC-MS

For each sample, a known quantity of TF or hemolymph was collected into a graduated glass capillary tube and blown into an individual glass vial containing 5 µL of 100% ethanol. Samples were kept at −20˚C until further processing. This biological sample was added to a 1:1 mixture of isooctane and methanol, vortexed for 30 s, and centrifuged for 7 min at maximum speed. Avoiding the boundary between phases, the majority of the isooctane layer and the methanol layer were removed separately, combined and stored at –80˚C until analysis. Before analysis, 50% acetonitrile (HPLC grade) was added. Prior to purification, farnesol (Sigma-Aldrich, St Louis, MO) was added to each sample to serve as an internal standard. Samples were extracted three times with hexane (HPLC grade). The hexane fractions were recombined in a clean borosilicate glass vial and dried by vacuum centrifugation. JH III was quantified using the gas chromatography mass spectrometry (GC–MS) method of *Bergot et al. (1981)* as modified by *Shu et al. (1997)* and *Brent and Vargo (2003)*. The residue was washed out of the vials with three rinses of hexane and added to borosilicate glass columns filled with aluminum oxide. In order to filter out contaminants, samples were eluted through the columns successively with hexane, 10% ethyl ether–hexane and 30% ethyl ether–hexane. After drying, samples were derivatized by heating at 60˚C for 20 min in a solution of methyl-d alcohol (Sigma-Aldrich) and trifluoroacetic acid (Sigma-Aldrich, St Louis, MO). Samples were dried down, resuspended in hexane, and again eluted through aluminum oxide columns. Non-derivatized components were removed with 30% ethyl ether. The JH derivative was collected into new vials by addition of 50% ethyl-acetate–hexane. After drying, the sample was resuspended in hexane. Samples were then analyzed using an HP 7890A Series GC (Agilent Technologies, Santa Clara, CA) equipped with a 30 m x 0.25 mm Zebron ZB-WAX column (Phenomenex, Torrence, CA) coupled to an HP 5975C inert mass selective detector. Helium was used as a carrier gas. JH form was confirmed by first running test samples in SCAN mode for known signatures of JH 0, JH I, JH II, JH III and JH III ethyl; JH III was confirmed as the primary endogenous form in this species. Subsequent samples were analyzed using the MS SIM mode, monitoring at m/z 76 and 225 to ensure specificity for the $d_3$-methoxyhydrin derivative of JH III. Total abundance was quantified against a standard curve of derivatized JH III, and adjusted for the starting volume of TF. The detection limit of the assay is approximately one pg.

## Long-term development

To determine the effect of exogenous JH on larval development, ants were taken from laboratory *C. floridanus* colonies (Expt 1: C2, C3, C5, C9, C16, C17; Expt 2: C4, C5, C6, C18; Expt 3: C1, C5, C6, C7, C11, C16, C19). Approximately 90% of the ants were taken from inside the nest on the brood, while the remaining 10% were taken from outside the nest. Each colony explant had 20–30 workers (each treatment had the same number of replicates of any given colony) and was provided with five to ten second or third instar larvae from their own colony of origin (staged larvae were equally distributed across replicates). Each explant was provided with water, and either solvent- or JH III-supplemented 30% sugar water and maple-syrup-based ant diet (1500 ng of JH III in 5 µL of ethanol was applied to each 3×3×3 mm food cube and sucrose solution had 83 ng JH III/µL). No insect-based food was provided. Food sources were refreshed twice per week. JH was found to transition

to JH acid gradually at room temperature, where after 1 week ~ 50% was JH acid (as measured by radio-assay as in *Kamita et al., 2011*, data not shown).

Twice weekly before feeding, each explant was checked for pupae, and developing larvae were measured and counts using a micrometer in the reticle of a stereomicroscope. Upon pupation, or cocoon spinning, larvae/pupae were removed from the care of workers and kept in a clean humid chamber until metamorphosis. Cocoons were removed using dissection forceps. The head width of the pupae was measured using a micrometer in the reticle of a stereomicroscope 1–4 days after metamorphosis (immediately after removal of the larval sheath, head width is not stable). Long-term development experiments were stopped when fewer than three larvae remained across all explants and these larvae had not changed in size for 2 weeks. Of larvae that did not successfully undergo metamorphosis, approximately 75% were eaten by nursing workers at varying developmental stages over the course of the experiment. The remaining non-surviving larvae were split between larvae that finished the cocoon spinning phase (*Wallis, 1960*) but did not complete metamorphosis and larvae that had ceased to grow by the end of the approximately 10-week experiment.

## *C. fellah* transcriptomics

### Transcriptome sequencing
We sequenced the *C. fellah* transcriptome (RNAseq) of workers from a single colony initiated from a queen collected during a mating flight in 2007 in Tel Aviv, Israel. Total RNA was extracted from the whole body of four minor workers with RNeasy Plus micro kit and RNase-Free DNase Set (QIAGEN, Hilden, Germany) and then 350 ng cDNA from each ant were pooled and sequenced. Illumina cDNA library was constructed and sequenced using strand-specific, paired-end sequencing of 100 bp reads. The library was sequenced on an Illumina HiSeq machine, which generated a total of 115 million pairs of reads.

### Transcriptome assembly
We ran Trinity (*Grabherr et al., 2011*) (version r2013-02–25, RRID:SCR_013048) on these sequence reads to assemble the *C. fellah* transcriptome. Reads were filtered according to the Illumina Chastity filter and then trimmed and filtered using Trimmomatic (*Bolger et al., 2014*) (RRID:SCR_011848, parameters: ILLUMINACLIP:TruSeq3-PE.fa:2:30:10 SLIDINGWINDOW:4:15 MINLEN:50). Trinity was run with default parameters for strand-specific, paired-read data. The assembled transcriptome consists of 66,156 genes ('components') with 99,402 transcripts, with putative open-reading frames found in 9987 and 27,062 of them, respectively. The total sequence length is 109,526,661 bases (including alternative splice variants) and the N50 size is 2243 bases. The *C. fellah* transcriptome dataset is available under accession number PRJNA339034.

## Protein orthology

In order to identify orthologous proteins across the species TF, we first needed to determine orthology across the four species. Compiling RefSeq, UniProt, and transcriptome protein models from the four species yielded 131,122 predicted protein sequences. For *C. floridanus*, *A. mellifera* and *S. invicta*, this was done for both NCBI RefSeq and UniProt databases because there are discrepancies in annotation and thus in protein identification between databases. We determined 21,836 groups of one-to-one orthologs using OMA stand-alone (*Altenhoff et al., 2015*) v.1.0.3, RRID:SCR_011978, although only 4538 orthologous groups had members in all four species. Default parameters were used with the exception of minimum sequence length, which was lowered to 30 aa.

The 40 proteins with the highest average NSAF value across TF samples of that species were selected. For each of the top proteins, we checked across the other species and databases for orthologous proteins. If an orthologous protein was identified in the proteome, we then checked if that protein was also present in that species' TF, even at low abundance.

## Dendrogram

The protein sequences of our proteomically identified TF *C. floridanus* JHE/Est-6 proteins, functionally validated JHEs (*Tribolium castaneum* JHE (UniProt D7US45), *A. mellifera* JHE (Q76LA5), *Manduca sexta* JHE (Q9GPG0), *D. melanogaster* JHE (A1ZA98) and *Culex quinquefasciatus* (R4HZP1)) and Est-6 proteins from *A. mellifera* (B2D0J5) and *D. melanogaster* (P08171) were aligned using

PROMALS3D with the crystal structure of the *M. sexta* JHE (2fj0) used as a guide. Phylogeny was inferred using Randomized Axelerated Maximum Likelihood (RAxML, RRID:SCR_006086; *Stamatakis, 2006*), with 100 bootstrapped trees. The dendrogram was visualized in FigTree (v1.4.2, RRID: SCR_008515).

## Sample sizes, data visualization and statistics

The proteomic sample sizes were determined by the variation observed in protein IDs and abundances in unmanipulated samples. Visualization together with hierarchical clustering was done in R version 3.0.2 (www.r-project.org, RRID:SCR_001905) using the 'heatmap.3' package in combination with the pvclust package. Heatmap visualization without clustering was done in MATLAB (2012b, RRID:SCR_001622). For experiments in *Figures 3* and *4*, the number of colonies and number of replicates or ants per colony were determined by both preliminary trials to assess sample variation and the health/abundance of the *C. floridanus* ant colonies available in the lab. Hydrocarbon GC-MS traces were analyzed in MATLAB normalizing the abundance (area under the curve) of each point by maximum and minimum value within the CHC retention time window. Peaks were found using 'findpeaks' with a lower threshold abundance of 7% of the total abundance for that sample. To compare across samples, peaks were filtering into 0.03 min bins by retention time. Cross-correlation was computed using the 'xcorr' function. For long-term development experiments, the number of same-staged larvae per colony was the limiting factor for the number of replicates per experiment. Because of this limitation, the long-term development experiment testing the effect of JH was fully repeated three times. Colonies were hibernated approximately 1 month prior to each of these experiments to maximize number of same-staged brood. General linear mixed models (GLMM) were used so that colony, replicate and experiment could be considered as random factors. Models were done in R using 'lmer' and 'glmer' functions of the lme4 package, and p values were calculated with the 'lmerTest' package in R. Overall, no data points were excluded as outliers and all replicates discussed are biological not technical replicates.

## Acknowledgements

We thank Robert I'Anson Price and Thomas Richardson for access to *A. mellifera* colonies, and Raphaël Braunschweig for access to *C. fellah* colonies. We thank George Shizuo Kamita for advice on JHEs. We would like to thank Sylviane Moss and Kay Harnish for help with high-throughput sequencing and Alexandra Bezler, Thomas Auer, Roman Arguello, Juan Alcaniz-Sanchez, Pavan Ramdya, Lucia Prieto-Godino, Katja Hoedjes, Raphaël Braunschweig and Sean McGregor for their useful comments on the manuscript. This work was funded by an ERC Advanced Grant (249375) and a Swiss NSF grant to LK, an ERC Starting Independent Researcher and Consolidator Grants (205202 and 615094) and a Swiss NSF Grant to R.B. Mention of trade names or commercial products in this article is solely for the purpose of providing specific information and does not imply recommendation or endorsement by the U.S. Department of Agriculture. USDA is an equal opportunity provider and employer. ZSG was supported by CAPES Brazil. EAM was supported by Wellcome Trust (104640/Z/14/Z).

## Additional information

### Funding

| Funder | Grant reference number | Author |
| --- | --- | --- |
| European Research Council | Advanced Grant 249375 | Laurent Keller |
| Schweizerischer Nationalfonds zur Förderung der Wissenschaftlichen Forschung | | Richard Benton Laurent Keller |
| European Research Council | Starting Independent Researcher 205202 | Richard Benton |
| European Research Council | Consolidator Grant 615094 | Richard Benton |

| Wellcome | Wellcome Trust grant 104640/Z/14/Z | Eric A Miska |
| Coordenação de Aperfeiçoamento de Pessoal de Nível Superior | | Zamira G Soares |

The funders had no role in study design, data collection and interpretation, or the decision to submit the work for publication.

## Author contributions

Adria C LeBoeuf, Managed the project and did all sample collection, isolation and long-term development experiments, data analysis, and figure production, Wrote the manuscript, Conception and design, Contributed unpublished essential data or reagents; Patrice Waridel, Performed proteomic LC-MS/MS for all samples; Colin S Brent, Performed GC-MS for Fig. 4A and B; Andre N Gonçalves, Performed small RNA bioinformatics analyses; Laure Menin, Daniel Ortiz, Performed GC-MS for Fig. 3 and analysis for molecule identification in Table 1; Oksana Riba-Grognuz, Performed expression analysis; Akiko Koto, Performed the RNA preparation and EP performed assembly and annotation of the *C. fellah* transcriptome, Contributed unpublished essential data or reagents; Zamira G Soares, Performed RNA extraction and library preparation for small RNA library; Eyal Privman, Performed assembly and annotation of the *C. fellah* transcriptome, Contributed unpublished essential data or reagents; Eric A Miska, Designed the small RNA sequencing experiments; Richard Benton, Laurent Keller, Oversaw and directed the project, Wrote the manuscript, Conception and design, Analysis and interpretation of data

## Author ORCIDs

Laurent Keller http://orcid.org/0000-0002-5046-9953

## Decision letter and Author response

Decision letter https://doi.org/10.7554/eLife.20375.030
Author response https://doi.org/10.7554/eLife.20375.031

# Additional files

## Supplementary files

• Supplementary file 1. RNA of microorganisms present in trophallactic fluid.
DOI: https://doi.org/10.7554/eLife.20375.020

• Supplementary file 2. Proteomic experiments included in this study.
DOI: https://doi.org/10.7554/eLife.20375.021

• Supplementary file 3. Orthology matrix across four social insect species.
DOI: https://doi.org/10.7554/eLife.20375.022

## Data availability

The following datasets were generated:

| Author(s) | Year | Dataset title | Dataset URL | Database and Identifier |
| --- | --- | --- | --- | --- |
| LeBoeuf AC, Waridel P, Brent CS, Gonçalves AN, Menin L, Ortiz D, Koto A, Privman E, Soares ZG, Miska EA, Benton R and Keller L | 2016 | Social exchange of chemical cues, growth proteins and hormones through trophallaxis | https://doi.org/10.6019/PXD004825 | ProteomeXchange, 10.6019/PXD004825 |
| LeBoeuf AC, Waridel P, Brent CS, Gonçalves AN, Menin L, Ortiz D, Koto A, Privman E, | 2016 | Camponotus fellah Transcriptome or Gene expression | http://www.ncbi.nlm.nih.gov/bioproject/PRJNA339034 | NCBI BioProject, PRJNA339034 |

| | | | | |
|---|---|---|---|---|
| Soares ZG, Miska EA, Benton R and Keller L | | | | |
| LeBoeuf AC, Waridel P, Brent CS, Gonçalves AN, Menin L, Ortiz D, Koto A, Privman E, Soares ZG, Miska EA, Benton R and Keller L | 2016 | Camponotus floridanus Transcriptome or Gene expression | http://www.ncbi.nlm.nih.gov/bioproject/PRJNA338939 | NCBI BioProject, PRJNA338939 |

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
