## [Decision Letter]

Thank you for submitting your article "Oral transfer of chemical cues, growth proteins and hormones in social insects" for consideration by *eLife*. Your article has been reviewed by two peer reviewers, and the evaluation has been overseen by a Reviewing Editor and Ian Baldwin as the Senior Editor. The following individual involved in review of your submission has agreed to reveal his identity: Markus Knaden (Reviewer #2).

The reviewers have discussed the reviews with one another and the Reviewing Editor has drafted this decision to help you prepare a revised submission.

The reviewers and Reviewing Editor were impressed by the amount of data collected in this study. Given that trophallaxis was considered to be a way of transferring specific sets of CHC in their opinion the main value of this study is that it provides an extensive analysis of not only CHCs, but also JH and JH-related compounds, proteins and micro-RNAs.

The fact that trophallaxis fluid contains a range of compounds including proteins and micro-RNAs is very interesting indeed and begs for studies to show what the effects of the components are. This study made some investigations of the effects of JH, a hormone known to affect insect physiology and a remarkable component in the TF. The other compounds are left untouched with the justification that genetic modification of ants is severely limited. Thus, the study does show that TF contains many potentially interesting compounds which may open studies by other researchers studying social insects.

This makes it potentially suitable for publication in *eLife*. However, the manuscript currently lacks sufficient discussion of the finding of JH plus JHE in the TF as well as a critical evaluation of why the ants transfer JH and how we should see the high levels in the TF in connection to the data on JH titer in 3rd instar larvae (not clear whether that is titer per larva or based on specific tissues).

Also, the information on the CHCs recorded should be presented in the manuscript. Finally, the proteomic data should be discussed in depth (what kind of proteins were found and what major inferences can be made on the basis of that? Also, the origin of the proteins deserves being discussed. The crop, from where the trophallactic fluid is presumably delivered is lined with a cuticular intima, being part of the foregut. This is not very permeable to large molecules such as proteins, which raises a dual question: How are these proteins sequestered (if at all) to the crop of the donor, and how they are transferred from the crop to the hemolymph of the recipient. If you feel that you can effectively address these issues – in addition to the other issues raised – we invite you to submit a revised version of your manuscript.

For the detailed comments by the reviewers, please see below.

*Reviewer #1:*

The manuscript encompasses massive amount of work, and the subject, that is the role of trophallaxis and the trophallactic fluid in social bonding is very important. However, despite the massive work done, the result is rather descriptive, in a way a catalog of what is in the fluid. While the information is important I am not sure whether it fits the scope of *eLife* or its general readers. Except for a small experiment of feeding JH, there is little attempt to examine the possible role or effect of the plethora of chemicals found in the trophallactic fluid. Neither have they attempted to address this question in the Discussion, which is surprisingly short considering the large amount of data presented.

Not being an expert in proteomics or genomics I do ask a question about the usefulness of the long list of proteins without giving any idea on how the operate or whether they have an effect. This is particularly important since the destination of the fluid is the other ant crop, or that of a larva in case of feeding). What happens to the proteins afterward is neither revealed nor discussed. This also goes for the microRNA.

I was surprised in a way about the presence of JH in the fluid, in particular given the fact that JH contain its degrading enzyme JH esterase. Again, there is no attempt to explain this phenomenon. I also doubt whether this JH may play a role in larval fate (size, length of the install etc.). Generally, due to its importance in many processes both in the larvae and adults, its titer is highly regulated. I cannot imagine that JH that reach the crop stays as is and transfer to the hemolymph. Generally it is bound to some kind of lipophorin that not only protects it from degradation but also facilitate its transport in the mostly hydrophilic milieu of the body fluids. Applying exogenous JH may have altered the developmental process of the larvae, but then it requires that workers control the level of JH in their crop if they wish to control larval size.

The presence of hydrocarbons in the trophallactic fluid is not surprising, as it was reported earlier. It is a pity on the other hand that the authors did not analyze also the postpharyngeal gland and only match the composition to the cuticle. Earlier studies showed that the amount in the crop is probably overflow from the PPG, rather than natural accumulation (Soroker and Hefetz 200 JIP 46:1097-112).

*Reviewer #2:*

In their manuscript "Oral transfer of chemical cues, growth proteins and hormones in social insects" LeBoeuf et al. show that, by conducting trophallaxis, ants (and bees) do not only share food, but potentially manipulate each other by the exchange of growth-related proteins and hormones. The authors furthermore show that (as suggested but never shown before) trophallaxis fluid contains cuticular hydrocarbons that are involved in forming the colony-specific olfactory/gustatory profile. This information is new and of high interest, as it brings us closer to the understanding on how e.g. the development of different castes or worker sizes is organised on the colony level. All results are discussed appropriately. The experiments are precisely designed and conducted carefully and it is admirable that the authors managed to milk fluid from freely trophalacting ants. Although the major part of the analysis was done with forced-milked fluids, the controls showing no difference between the two milking strategies are convincing. Being no chemist I appreciate how well the chemical results (regarding presence or absence, or specific amounts of numerous identified substances) are presented in the figures. I fully support the publication of this manuscript in *eLife*.

---

## [Author Response]

[…] This makes it potentially suitable for publication in eLife. However, the manuscript currently lacks sufficient discussion of the finding of JH plus JHE in the TF.

We have addressed this in the penultimate paragraph of the Discussion, proposing two potential models, one of negative regulation/buffering of JH by JHE and one of neofunctionalization of JHE.

Excerpt from text: “While the simultaneous presence of JH and a set of putative JH degrading enzymes (the JHE/Est-6 family) in *C. floridanus* TF may appear surprising, there are at least two possible explanations. […] *Drosophila* Est-6 is highly abundant in seminal fluid, together with its presumed substrate, the male-specific pheromone 11-cis-vaccenyl acetate (cVA). cVA has multiple roles in influencing sexual and aggressive interactions and its transfer to females during copulation potently diminishes her attractiveness to future potential mates [1,104,105].”

As well as a critical evaluation of why the ants transfer JH and how we should see the high levels in the TF in connection to the data on JH titer in 3rd instar larvae (not clear whether that is titer per larva or based on specific tissues).

We have strengthened the Results section to clarify how the JH titer measurements were performed in larvae, and now detail our reasoning for why the TF contains JH in the third paragraph of the Discussion.

Excerpt from Results: “Because JH is an important regulator of development, reproduction, and behavior [Nijhout and Wheeler, 1982], we investigated whether the dose of JH that a larva receives by trophallaxis with a nursing worker might be physiologically relevant. […] While it is difficult to determine how much of the JH fed to larvae remains in the larval digestive tract, these results indicate that there is potentially sufficient JH in a single trophallaxis-mediated feeding to shift the titer of a recipient larva.”

Excerpt from Discussion: “The presence of development- and growth-related components in the TF of diverse social insects suggests that this fluid may play a role in directing larval development. […] The between-genera variation in the most abundant growth-related proteins (e.g., MRJPs in honey bee, JH-binding protein and hexamerins in fire ant, and JHEs in Camponotus) indicates that there might be multiple evolutionary origins and/or rapid divergence in trophallaxis-based signals potentially influencing larval growth.”

Also, the information on the CHCs recorded should be presented in the manuscript.

We have now included a list of the molecules in TF that we were able to identify by GC-MS (Table 1), and made corresponding additions to the Results, Legends and Methods sections.

Excerpt from Results: “We identified 63 molecules in TF (Table 1), including eight fatty acids and fatty acid esters, 13 unbranched and 36 branched hydrocarbons with one to five methyl branches. […] Altogether, these observations indicate that while CHCs are likely to be exchanged by trophallaxis, additional mechanisms are probably involved in generating the colony-specific bouquet of these compounds.”

Finally, the proteomic data should be discussed in depth (what kind of proteins were found and what major inferences can be made on the basis of that? Also, the origin of the proteins deserves being discussed. The crop, from where the trophallactic fluid is presumably delivered is lined with a cuticular intima, being part of the foregut. This is not very permeable to large molecules such as proteins, which raises a dual question: How are these proteins sequestered (if at all) to the crop of the donor, and how they are transferred from the crop to the hemolymph of the recipient.

We agree that this important information should be provided. We therefore performed three new analyses to address these questions:

1) To provide insight into the site of expression of the TF proteins, we have re-analyzed existing transcriptomic (RNA-seq) data from *C. floridanus* brain and head+thorax samples (Simola et al. 2013). These analyses indicate that many TF proteins are highly expressed in the head and thorax but less so in the brain (Author response image 1). However, because of the limited diversity and resolution of available tissue-specific RNA-seq data in *C. floridanus*, and because there is considerable variation in the data from brains (possible because of contamination during dissections), we feel that it is better not to include these data in the manuscript itself.

**Author response image 1. respfig1:** Heatmap comparing TF protein abundance with RNA expression of the same genes in brain, head and thorax samples from RNA-Seq datasets (Simola et al. Genome Research 2013). Expression values are calculated as transcripts per million (TPM) reads, and normalized for presentation on the same scale as proteomic data from Figure 5 (NSAF).

2) To examine how TF protein gets into the crop, we have analyzed the protein sequences for the presence/absence of secretion signal peptides (Figure 5—source data 1). As expected, the large majority of proteins have predicted signal sequences.

3) Although the functions of almost all *C. floridanus* TF proteins (and those of other social insects analyzed) have not been directly assessed, we have identified *D. melanogaster* orthologs and incorporated available expression/localization/functional data for this species’ proteins into the same orthology table (Figure 5—source data 1). Half of the *D. melanogaster* orthologous proteins lacking a secretion signal had gene ontology terms indicating extracellular or lipid-particle localization suggesting that they may gain access to the lumen of the foregut through other protein secretion pathways. While it is premature to speculate too deeply about the roles of individual TF proteins, we have presented in more detail some of the main functional groups in the Discussion, and hope that this compiled data will be a helpful resource for follow-up studies.

Excerpt from Results on transport: “Thirty-three of the 50 most abundant TF proteins had predicted N-terminal signal peptides, suggesting they can be secreted directly into this fluid by cells lining the lumen or glands connected to the alimentary canal (Figure 5—source data 1). Moreover, half of the proteins without such a secretion signal had gene ontology terms indicating extracellular or lipid-particle localization, which suggests that they may gain access to the lumen of the foregut through other transport pathways.”

Excerpt from Discussion on immune defense: “Consistent with the view that the gut is one of the first lines of defense in the body’s interface with the outside world (Lemaitre and Hoffmann, 2007; Söderhäand Cerenius, 1998), the four TF proteomes analyzed in this study include many potential defense-related proteins. […] Their presence may simply reflect a few cells being sloughed from the lumen of the foregut, or could be indicative of a defense process termed ETosis, whereby chromatin is released from the nuclei of inflammatory cells to form extracellular traps that kill pathogenic microbes (Robb et al., 2014; Brinkmann et al., 2004).”

Excerpt from Discussion on transport: “The finding that TF contains many proteins, miRNAs, CHCs and JH raises the question of how they come to be present in this fluid. […] Extracellular miRNAs are secreted and transported through a variety of pathways, but the functional relevance of such molecules is still controversial (Sarkies and Miska, 2013; Turchinovich, Tonevitsky and Burwinkel, 2016; Masood et al., 2016; Søvik, Bloch and Ben-Shahar, 2015; Rayner and Hennessy, 2013).

For the detailed comments by the reviewers, please see below.Reviewer #1:[…] Not being an expert in proteomics or genomics I do ask a question about the usefulness of the long list of proteins without giving any idea on how the operate or whether they have an effect. This is particularly important since the destination of the fluid is the other ant crop, or that of a larva in case of feeding). What happens to the proteins afterward is neither revealed nor discussed. This also goes for the microRNA.

We acknowledge that only a few experiments were performed to manipulate the role or effect of the plethora of chemicals found in the trophallactic fluid. Such experiments are unfortunately difficult to conduct in ant colonies with current tools. We decided to manipulate JH because it is a molecule that is likely to be the regulatory target of some of the more-difficult-to-manipulate proteins in the TF.

I was surprised in a way about the presence of JH in the fluid, in particular given the fact that JH contain its degrading enzyme JH esterase. Again, there is no attempt to explain this phenomenon. I also doubt whether this JH may play a role in larval fate (size, length of the install etc.). Generally, due to its importance in many processes both in the larvae and adults, its titer is highly regulated. I cannot imagine that JH that reach the crop stays as is and transfer to the hemolymph. Generally it is bound to some kind of lipophorin that not only protects it from degradation but also facilitate its transport in the mostly hydrophilic milieu of the body fluids. Applying exogenous JH may have altered the developmental process of the larvae, but then it requires that workers control the level of JH in their crop if they wish to control larval size.

The reviewer is correct regarding JH and lipophorins, and indeed, there are lipophorins in TF, making them obvious candidates for the transfer of JH into TF. Unfortunately, little is known on how JH gains access to tissues from the hemolymph or elsewhere, but given that many experiments apply this hormone (or its analogs) topically, JH appears to be able to cross cuticle. We have now extended the Discussion to address this issue.

Excerpt from Discussion: “The finding that TF contains many proteins, miRNAs, CHCs and JH raises the question of how they come to be present in this fluid. […] Extracellular miRNAs are secreted and transported through a variety of pathways, but the functional relevance of such molecules is still controversial (Sarkies and Miska, 2013; Turchinovich, Tonevitsky and Burwinkel, 2016; Masood et al., 2016; Søvik, Bloch and Ben-Shahar, 2015; Rayner and Hennessy, 2013).”

The presence of hydrocarbons in the trophallactic fluid is not surprising, as it was reported earlier. It is a pity on the other hand that the authors did not analyze also the postpharyngeal gland and only match the composition to the cuticle. Earlier studies showed that the amount in the crop is probably overflow from the PPG, rather than natural accumulation (Soroker and Hefetz 200 JIP 46:1097-112).

The goal of our current study was to examine what is in TF, rather than identify the source for these molecules. Nevertheless, to address the reviewer’s comment we have now included information about the PPG in the Discussion.